# SELF-SPECULATIVE MASKED DIFFUSIONS

**Andrew Campbell,**[*] **Valentin De Bortoli, Jiaxin Shi,**[†] **Arnaud Doucet**
Google DeepMind
andrew.campbell.ml@outlook.com
{vdebortoli, arnauddoucet}@google.com
ishijiaxin@gmail.com

## ABSTRACT

We present self-speculative masked diffusions, a new class of masked diffusion generative models for discrete data that require significantly fewer function evaluations to generate samples. Standard masked diffusion models predict factorized logits over currently masked positions. A number of masked positions are then sampled, however, the factorization approximation means that sampling too many positions in one go leads to poor sample quality. As a result, many simulation steps and therefore neural network function evaluations are required to generate high-quality data. We reduce the computational burden by generating *non-factorized* predictions over masked positions. This is achieved by modifying the final transformer attention mask from non-causal to causal, enabling draft token generation and parallel validation via a novel, model-integrated speculative sampling mechanism. This results in a non-factorized predictive distribution over masked positions in a single forward pass. We apply our method to GPT2 scale text modelling and protein sequence generation, finding that we can achieve a $\sim 2\times$ reduction in the required number of network forward passes relative to standard masked diffusion models.

## 1 INTRODUCTION

Generative models for discrete data are at the core of a wide range of modern deep learning systems from chatbots (Gemini Team, 2023; OpenAI Team, 2023; Anthropic, 2024) to biological foundation models (Hayes et al., 2025; Wang et al., 2025; Madani et al., 2023). These models generate data through an iterative process, each step revealing only a small portion of the final tokens. Autoregressive (AR) models (Sutskever et al., 2014; Brown et al., 2020) reveal a single token in each generation step, doing so in a left-to-right ordering. Alternatively, masked diffusion models (MDMs) and any-order AR models (Hoogeboom et al., 2022; Shih et al., 2022; Sahoo et al., 2024; Shi et al., 2024; Ou et al., 2025) reveal multiple tokens at each step and can operate using any given generation ordering. The ordering flexibility is beneficial for applications without an inherent left-to-right structure such as protein sequences (Alamdari et al., 2023; Gruver et al., 2024; Wang et al., 2024).

During an update, standard MDMs use a neural network to output a factorized predictive distribution over currently masked positions. Revealing more than one token from this distribution incurs approximation error, as data distributions typically do not factorize that way; see e.g. (Zheng et al., 2025). This imposes an upper limit on how many tokens can be revealed per step without loss of quality, meaning a large number of network forward passes are required to generate a full datapoint.

Our objective is to reveal multiple masked tokens concurrently using a non-factorized predictive distribution. Inspired by self-speculative sampling (Zhang et al., 2024; Elhoushi et al., 2024; Liu et al., 2024), we propose generating a draft sequence using a network subset, then validating it in parallel with the full transformer. By employing speculative sampling (Leviathan et al., 2023; Chen et al., 2023), we ensure the accepted token sequence is distributed according to a *non-factorized* target distribution defined by the full capacity network.

Using self-speculative sampling for MDMs provides a non-factorized predictive distribution that can be sampled efficiently, but requires solving two key challenges. Firstly, speculative verification

---

[*]Work done during a Google DeepMind internship
[†]Now at Meta Superintelligence Labs

requires a causal transformer but standard MDMs use a non-causal transformer. We solve this with a novel hybrid non-causal and permutation informed causal architecture (Pannatier et al., 2024), building in residual connections so that the causal target distribution learns to improve over the non-causal draft distribution. Second, in contrast to standard speculative sampling (Leviathan et al., 2023), the non-causal layers create a target distribution which depends on the speculative sampling accept/reject sequence. We theoretically characterize this dependence by deriving a log-likelihood lower bound for this model class.

We demonstrate our method on GPT2-scale text modelling and protein sequence generation. On the OpenWebText and UniRef50 datasets, we achieve a $\sim 2\times$ reduction in the number of function evaluations (NFE) required for a given sample quality relative to standard MDMs.

## 2 BACKGROUND

### 2.1 MASKED DIFFUSIONS AS ANY-ORDER AUTOREGRESSIVE MODELS

We assume our data $\boldsymbol{x} = (\boldsymbol{x}^1, ..., \boldsymbol{x}^D)$ has $D$ dimensions and $S$ possible categories, i.e. $\boldsymbol{x} \in \{1, \ldots, S\}^D$. Given the values of $\boldsymbol{x}$ on some subset of these dimensions, masking style generative models train a neural network to predict the remaining unobserved dimensions. This can be expressed mathematically in two ways. The first, MDMs, define a continuous-time discrete-valued Markov chain $(\boldsymbol{x}_t^{1:D})_{t \in [0,1]}$ on the state-space $\{1, \ldots, S, S+1\}^D$, where an additional mask token $M$, $M := S + 1$, is introduced; see e.g. (Ou et al., 2025; Sahoo et al., 2024; Shi et al., 2024) for details. This Markov chain, initialized at $\boldsymbol{x}_0^d = \boldsymbol{x}^d$, is such that

$$p(\boldsymbol{x}_t^d | \boldsymbol{x}^d) = (1 - \alpha_t)\delta\{\boldsymbol{x}_t^d = \boldsymbol{x}^d\} + \alpha_t \delta\{\boldsymbol{x}_t^d = M\}, \tag{1}$$

for a non-decreasing noising schedule $\alpha_t \in [0,1]$ such that $\alpha_0 = 0$, $\alpha_1 = 1$. The time-reversal of this noising process defines a generative model consisting of $D$ random events spaced between $t = 0$ and $t = 1$, with each event revealing a single masked token. Individually simulating these $D$ events is computationally prohibitive when $D$ is large. Hence, many inference techniques instead reveal multiple masked tokens simultaneously using a factorized predictive distribution over the masked dimensions given the current partially masked context $\boldsymbol{x}_t^{1:D}$. This is parameterized by a neural network with parameters $\theta$ as,

$$p_\theta(\boldsymbol{x}^{1:D} | \boldsymbol{x}_t^{1:D}) = \prod_{d:\boldsymbol{x}_t^d = M} p_\theta(\boldsymbol{x}^d | \boldsymbol{x}_t^{1:D}) \prod_{d:\boldsymbol{x}_t^d \neq M} \delta\{\boldsymbol{x}^d = \boldsymbol{x}_t^d\}. \tag{2}$$

As shown by Austin et al. (2021); Hoogeboom et al. (2022); Ou et al. (2025), MDMs are closely related to any-order AR models which directly consider the unmasking events, doing away with the time variable $t$. Let $\sigma$ be a permutation of the numbers $\{1, \ldots, D\}$, with $\sigma(i : j) = (\sigma(i), \sigma(i+1), ..., \sigma(j))$ representing the $i$-th to the $j$-th element of this permutation. This permutation is usually picked uniformly at random. Given some subset of the dimensions of $\boldsymbol{x}$, any-order AR models predict the remaining dimensions by parameterizing the predictive distribution $p_\theta(\boldsymbol{x}^{\sigma(i+1:D)} | \boldsymbol{x}^{\sigma(1:i)})$. AR diffusion models (Hoogeboom et al., 2022) use a factorized parameterization of this distribution similar to MDMs

$$\overleftrightarrow{p}_\theta(\boldsymbol{x}^{\sigma(i+1:D)} | \boldsymbol{x}^{\sigma(1:i)}) = \prod_{d=i+1}^{D} \overleftrightarrow{p}_\theta(\boldsymbol{x}^{\sigma(d)} | \boldsymbol{x}^{\sigma(1:i)}), \tag{3}$$

where the $\overleftrightarrow{p}$ notation signifies $\overleftrightarrow{p}_\theta(\boldsymbol{x}^{\sigma(i+1:D)} | \boldsymbol{x}^{\sigma(1:i)})$ is parameterized with a transformer using a non-causal ($\leftrightarrow$) any-to-any attention mask. Conditioning on only the $\boldsymbol{x}^{\sigma(1:i)}$ dimensions is achieved by masking out the $\boldsymbol{x}^{\sigma(i+1:D)}$ dimensions at the input to the transformer. The distribution is *factorized* over the components of $\boldsymbol{x}^{\sigma(i+1:D)}$. We refer to $\overleftrightarrow{p}$ as a *non-causal* distribution.

An alternative approach is to parameterize the predictive distribution autoregressively, using a network architecture that can operate over a given input ordering (Yang et al., 2019; Pannatier et al., 2024), i.e.

$$\overrightarrow{p}_\phi(\boldsymbol{x}^{\sigma(i+1:D)} | \boldsymbol{x}^{\sigma(1:i)}) = \prod_{d=i+1}^{D} \overrightarrow{p}_\phi(\boldsymbol{x}^{\sigma(d)} | \boldsymbol{x}^{\sigma(1:d-1)}), \tag{4}$$

where the $\overrightarrow{p}$ notation signifies that $\overrightarrow{p}_\phi(\boldsymbol{x}^{\sigma(i+1:D)} | \boldsymbol{x}^{\sigma(1:i)})$ is parameterized with a transformer using a causal ($\rightarrow$) attention mask to ensure it only attends to $\boldsymbol{x}^{\sigma(1:d-1)}$. We provide an overview on

---

**Algorithm 1** Mask Generative Model Sampling With Non-Causal $\overset{\leftrightarrow}{p}_\theta$

---

$i \leftarrow 0$    Number of tokens currently revealed
$\sigma \sim p(\sigma)$    Draw the generation ordering
**while** $i < D$ **do**
   $k \sim p(k|i)$    Number of tokens to reveal, $k \geq 1$
   $\boldsymbol{x}^{\sigma(i+1:i+k)} \sim \overset{\leftrightarrow}{p}_\theta(\boldsymbol{x}^{\sigma(i+1:i+k)}|\boldsymbol{x}^{\sigma(1:i)})$
   $i \leftarrow i + k$
**end while**

---

attention masks in Appendix A. The distribution is *autoregressive* over the components of $\boldsymbol{x}^{\sigma(i+1:D)}$. We refer to $\overset{\rightarrow}{p}$ as a *causal* distribution.

Mask generative models with non-causal $\overset{\leftrightarrow}{p}_\theta$ are sampled using the multi-step process in Algorithm 1. Firstly, a generation ordering $\sigma$ is drawn from $p(\sigma)$. This step is explicit for any-order AR models whereas for standard MDMs $\sigma$ is implicitly picked uniformly at random. Then, within each update step, a number of tokens $k$ are revealed in the ordering specified by $\sigma$. Any-order AR models typically set $k$ to be some constant number of reveals per step, whereas standard MDMs reveal a random number of tokens per step. The number of reveals, $k$, is drawn from some distribution $p(k|i)$ that can be derived from the noise schedule $\alpha_t$ and depends on the current number of revealed tokens, $i$.

## 2.2 SPECULATIVE SAMPLING

Speculative sampling is a method developed to accelerate inference for large language models (LLMs) (Chen et al., 2023; Leviathan et al., 2023). The total cost of LLM inference is memory-bound, i.e., dominated by the cost of moving the model weights from high-bandwidth memory to the accelerator cache. Since cost is dominated by memory operations, applying the transformer to a sequence has similar latency to a single token, despite increased arithmetic computations in the former case. Speculative sampling exploits this by creating a draft sequence using a smaller model, which is then efficiently verified in parallel with a single forward pass of the larger target LLM. Verification results in the acceptance of a subset of the draft tokens, such that the accepted sequence is still distributed according to the probability distribution defined by the target model.

Formally, let the draft model be $p(\boldsymbol{x}^d|\boldsymbol{x}^{1:d-1})$ and the target model be $q(\boldsymbol{x}^d|\boldsymbol{x}^{1:d-1})$. We aim to draw samples from $q$ by first sampling a sequence of tokens from $p$ and accepting some number of them. We let the context of already generated tokens be $\boldsymbol{x}^{1:i}$ and let $W \geq 1$ be some window size. To sample new tokens, we first draw the draft sequence, $\hat{\boldsymbol{x}}^{i+1:i+W} \sim \prod_{d=i+1}^{i+W} p(\hat{\boldsymbol{x}}^d|\boldsymbol{x}^{1:i}, \hat{\boldsymbol{x}}^{i+1:d-1})$. We then, in parallel, obtain the probabilities that the target model assigns to the draft tokens $\hat{\boldsymbol{x}}^{i+1:i+W}$, $\{q(\hat{\boldsymbol{x}}^d|\boldsymbol{x}^{1:i}, \hat{\boldsymbol{x}}^{i+1:d-1})\}_{d=i+1}^{i+W}$. Starting from the $(i+1)$-th token, we accept each token with probability $\min\left(1, q(\hat{\boldsymbol{x}}^d|\boldsymbol{x}^{1:i}, \hat{\boldsymbol{x}}^{i+1:d-1})/p(\hat{\boldsymbol{x}}^d|\boldsymbol{x}^{1:i}, \hat{\boldsymbol{x}}^{i+1:d-1})\right)$. Let the first position we reject be $j$. We then resample $\hat{\boldsymbol{x}}^j$ from the adjusted distribution $r(\hat{\boldsymbol{x}}^j|\boldsymbol{x}^{1:i}, \hat{\boldsymbol{x}}^{i+1:j-1}) \propto \max\left(0, q(\hat{\boldsymbol{x}}^j|\boldsymbol{x}^{1:i}, \hat{\boldsymbol{x}}^{i+1:j-1}) - p(\hat{\boldsymbol{x}}^j|\boldsymbol{x}^{1:i}, \hat{\boldsymbol{x}}^{i+1:j-1})\right)$. One can prove that the resulting sequence $\hat{\boldsymbol{x}}^{i+1:j}$ is distributed according to the target distribution $q(\hat{\boldsymbol{x}}^{i+1:j}|\boldsymbol{x}^{1:i})$ (Leviathan et al., 2023).

## 3 SELF-SPECULATIVE MASKED DIFFUSIONS

To sample from MDMs efficiently, we want to reveal many tokens $k$ per update step (Algorithm 1). The standard approach computes the factorized distribution $\overset{\leftrightarrow}{p}_\theta(\boldsymbol{x}^{\sigma(i+1:D)}|\boldsymbol{x}^{\sigma(1:i)})$ (Equation 3) using one single forward pass and samples $k$ tokens from it. If $k$ is large, sample quality degrades due to the conditional independence approximation. Ideally, we would sample from the non-factorized distribution $\overset{\rightarrow}{p}_\phi(\boldsymbol{x}^{\sigma(i+1:D)}|\boldsymbol{x}^{\sigma(1:i)})$ (Equation 4). However, naively sampling $k$ tokens autoregressively from $\overset{\rightarrow}{p}_\phi$ requires $k$ forward passes, negating any computational savings.

To circumvent this, we use speculative sampling to efficiently sample from $\overset{\rightarrow}{p}_\phi(\boldsymbol{x}^{\sigma(i+1:D)}|\boldsymbol{x}^{\sigma(1:i)})$, using the non-causal model as the draft and the causal model as the target. To avoid needing two models, Section 3.1 describes a novel hybrid transformer with both non-causal and causal layers, enabling drafting and verification in one forward pass. We then describe our training objective

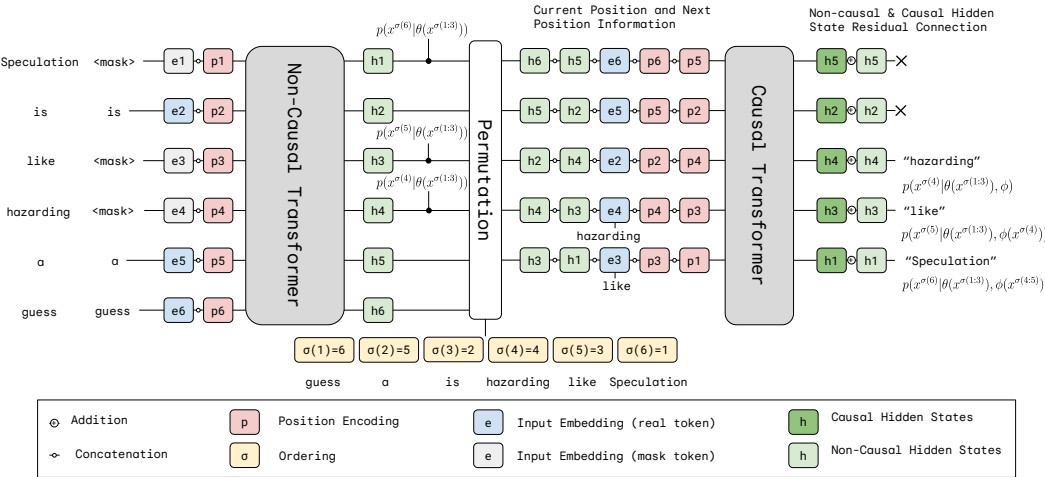

Figure 1: Hybrid non-causal/causal transformer during training on the sentence *"Speculation is like hazarding a guess"* which is partially corrupted in the ordering given by $\sigma(1:6) = [6, 5, 2, 4, 3, 1]$. We always mask the final elements in the sequence, here the final 3 elements in the sequence, $[4, 3, 1]$. At inference time, the causal transformer uses draft tokens rather than real tokens. Empty circles represent concatenation whilst circles with a plus represent addition.

(Section 3.2) and sampling algorithm (Section 3.3). This combined architecture induces a shifting speculative sampling target, which we characterize theoretically in Section 3.4, and discuss practical sampling improvements in Section 3.5.

### 3.1 ARCHITECTURE

We need to parameterize both a non-causal factorized distribution $\overset{\leftrightarrow}{p}_\theta(\boldsymbol{x}^{\sigma(i+1:D)}|\boldsymbol{x}^{\sigma(1:i)})$ and a causal AR distribution $\overset{\rightarrow}{p}_\phi(\boldsymbol{x}^{\sigma(i+1:D)}|\boldsymbol{x}^{\sigma(1:i)})$ for any permutation $\sigma$. We propose a hybrid architecture that starts with a set of non-causal blocks parameterizing $\overset{\leftrightarrow}{p}_\theta$ similar to standard MDMs. We then use a set of causal blocks operating on the hidden states from the previous layers to parameterize $\overset{\rightarrow}{p}_\phi$. These causal blocks follow the $\sigma$-GPT architecture (Pannatier et al., 2024), which models any ordering $\sigma$ by operating on the permuted sequence with additional positional encodings. Our full architecture is given in Figure 1, we now detail each component.

**Non-Causal Blocks.** These layers follow the standard MDM architecture: masked/unmasked tokens and positional encodings are passed into any-to-any attention layers. The factorized distribution $\overset{\leftrightarrow}{p}_\theta(\boldsymbol{x}^{\sigma(i+1:D)}|\theta(\boldsymbol{x}^{\sigma(1:i)}))$ is computed from the output hidden states; the track corresponding to $\sigma(j)$ predicting the token in position $\sigma(j)$ for $j > i$. For example, in Figure 1, $\overset{\leftrightarrow}{p}_\theta(\boldsymbol{x}^{\sigma(5)}|\theta(\boldsymbol{x}^{\sigma(1:3)}))$ is computed from the hidden state on the track corresponding to the fifth position in the ordering (*h3* in this case since $\sigma(5) = 3$). The notation $\theta(\boldsymbol{x}^{\sigma(1:i)})$ denotes that tokens $\boldsymbol{x}^{\sigma(1:i)}$ were revealed as a conditioning signal that was subsequently processed by the non-causal layers with parameters $\theta$.

**Causal Blocks.** We parameterize the causal distribution via the $\sigma$-GPT architecture: left-to-right attention masked causal blocks applied to the permuted sequence. Unlike non-causal blocks, the track corresponding to position $\sigma(j)$ predicts the token for the next position in the sequence, $\sigma(j+1)$. This is standard for causally masked transformers; since the full unmasked sequence is input, predicting the token $\sigma(j)$ from track $\sigma(j)$ would be trivial. Because we operate on any ordering $\sigma$, each track needs information on both its own position, $\sigma(j)$, and the next position $\sigma(j+1)$, provided via positional encodings (Pannatier et al., 2024). For example, in Figure 1, the $\sigma(5)$ track uses positional encodings for $\sigma(5) = 3$ and $\sigma(6) = 1$ to predict the token *"Speculation"* in position $\sigma(6) = 1$

In our architecture, we also input the non-causal hidden states corresponding to the current and next position into the causal blocks, following the same reasoning as for the positional encodings. For the first position, $\sigma(1)$, we set the causal distribution equal to the non-causal distribution as no extra information is available.

**Output.** At the causal output, for the prediction of token in position $\sigma(j+1)$, we use a residual connection adding the non-causal hidden state for position $\sigma(j+1)$ e.g. *h1* for the token *"Speculation"* in Figure 1. This ensures the causal target learns to improve over the non-causal distribution. It also aligns the draft and target distributions, increasing the speculative acceptance rate. The causal output from our hybrid architecture depends on both the parameters of the non-causal blocks ($\theta$) and the causal blocks ($\phi$). We write this as $\overrightarrow{p}_{\theta,\phi}(\boldsymbol{x}^{\sigma(d)}|\theta(\boldsymbol{x}^{\sigma(1:i)}),\phi(\boldsymbol{x}^{\sigma(i+1:d-1)}))$ specifying that tokens $\boldsymbol{x}^{\sigma(1:i)}$ were processed by $\theta$ in the non-causal blocks, whilst future tokens $\phi(\boldsymbol{x}^{\sigma(i+1:d-1)})$ were input only into the causal blocks.

**Training and Inference.** During training, we take a true token sequence, apply masks and input this into the non-causal blocks with the model predicting the true token values in the masked positions. We input the permuted full true token sequence into the causal blocks without masks, with the causal attention operation ensuring each track can only attend to tokens prior to it in the permuted sequence.

During inference, let the currently revealed set of tokens be $\boldsymbol{x}^{\sigma(1:i)}$. We pass $\boldsymbol{x}^{\sigma(1:i)}$ into the non-causal blocks, with any positions not yet revealed being represented with a mask token. For the causal blocks, which take a full real token sequence as input (no mask tokens), we pass $\boldsymbol{x}^{\sigma(1:i)}$ as well as draft tokens generated by the non-causal blocks for future positions, as we will see in Section 3.3.

## 3.2 TRAINING OBJECTIVE

When the revealed context to the non-causal blocks is $\boldsymbol{x}^{\sigma(1:i)}$, our architecture parameterizes both the non-causal distribution

$$\overleftrightarrow{p}_{\theta}(\boldsymbol{x}^{\sigma(i+1:D)}|\theta(\boldsymbol{x}^{\sigma(1:i)})) = \prod_{d=i+1}^{D} \overleftrightarrow{p}_{\theta}(\boldsymbol{x}^{\sigma(d)}|\theta(\boldsymbol{x}^{\sigma(1:i)})), \tag{5}$$

and the causal distribution

$$\overrightarrow{p}_{\theta,\phi}(\boldsymbol{x}^{\sigma(i+1:D)}|\theta(\boldsymbol{x}^{\sigma(1:i)}),\phi) = \prod_{d=i+1}^{D} \overrightarrow{p}_{\theta,\phi}(\boldsymbol{x}^{\sigma(d)}|\theta(\boldsymbol{x}^{\sigma(1:i)}),\phi(\boldsymbol{x}^{\sigma(i+1:d-1)})). \tag{6}$$

We would like both the non-causal and causal distributions to approximate the corresponding true conditional distributions of the data distribution

$$\overleftrightarrow{p}_{\theta}(\boldsymbol{x}^{\sigma(d)}|\theta(\boldsymbol{x}^{\sigma(1:i)})) \approx p_{\text{data}}(\boldsymbol{x}^{\sigma(d)}|\boldsymbol{x}^{\sigma(1:i)}) \tag{7}$$

$$\overrightarrow{p}_{\theta,\phi}(\boldsymbol{x}^{\sigma(d)}|\theta(\boldsymbol{x}^{\sigma(1:i)}),\phi(\boldsymbol{x}^{\sigma(i+1:d-1)})) \approx p_{\text{data}}(\boldsymbol{x}^{\sigma(d)}|\boldsymbol{x}^{\sigma(1:d-1)}). \tag{8}$$

We note that in Equation 8, there are multiple causal distributions that approximate the same true conditional $p_{\text{data}}(\boldsymbol{x}^{\sigma(d)}|\boldsymbol{x}^{\sigma(1:d-1)})$ for different values of $i$ which specifies the computation path taken to parameterize $\overrightarrow{p}_{\theta,\phi}$. We discuss this further in Section 3.3. We train $\theta$ and $\phi$ by jointly maximizing the standard cross entropy loss with a weighting $\frac{D}{D-i}$ that normalizes by the number of masked positions $D-i$ (Hoogeboom et al., 2022)

$$\mathcal{L} = \mathbb{E}\left[\frac{D}{D-i}\left(\log \overleftrightarrow{p}_{\theta}(\boldsymbol{x}^{\sigma(i+1:D)}|\theta(\boldsymbol{x}^{\sigma(1:i)})) + \log \overrightarrow{p}_{\theta,\phi}(\boldsymbol{x}^{\sigma(i+1:D)}|\theta(\boldsymbol{x}^{\sigma(1:i)}),\phi)\right)\right], \tag{9}$$

where the expectation is taken over $p_{\text{data}}(\boldsymbol{x})p(\sigma)p(i)$ with $\sigma$ sampled uniformly and $p(i)$ following a noising schedule similar to MDMs with $p(i=D)=0$. The first part of the loss is mathematically equivalent to the MDM loss (Shi et al., 2024; Sahoo et al., 2024), as shown by Ou et al. (2025). The second part of the loss can be seen as a standard AR cross-entropy loss computed over the masked tokens under a random order (Uria et al., 2014). We highlight that evaluating both $\overleftrightarrow{p}_{\theta}$ and $\overrightarrow{p}_{\theta,\phi}$ for the objective takes only one single forward pass of the hybrid network. In Appendix E we carry out a FLOP analysis for the extra overhead in our architecture versus a standard transformer, finding only a $0.98\%$ increase in FLOPs for standard settings.

Training curves (Section 5.1) show that the causal distribution $\overrightarrow{p}_{\theta,\phi}$ achieves a significantly lower loss than the non-causal draft distribution $\overleftrightarrow{p}_{\theta}$, as it can model missing tokens with a non-factorized distribution. We note that our architecture is compatible with both fine-tuning and full-training. In fine-tuning we can freeze the non-causal backbone to a pretrained masked diffusion model and train only the causal blocks on top, as we demonstrate in Section 5.3.

---

**Algorithm 2** Self-Speculative Masked Diffusion Sampling

$i \leftarrow 0$    Number of tokens currently revealed
$\sigma \sim p(\sigma)$    Draw the generation ordering
**while** $i < D$ **do**
  $\hat{\boldsymbol{x}}^{\sigma(i+1:D)} \sim \overleftrightarrow{p}_\theta(\hat{\boldsymbol{x}}^{\sigma(i+1:D)}|\theta(\boldsymbol{x}^{\sigma(1:i)}))$    Sample draft tokens
  Compute   $\{\overrightarrow{p}_{\theta,\phi}(\hat{\boldsymbol{x}}^{\sigma(d)}|\theta(\boldsymbol{x}^{\sigma(1:i)}), \phi(\hat{\boldsymbol{x}}^{\sigma(i+1:d-1)}))\}_{d=i+1}^D$
  **for** $d = i+1$ to $D$ **do**
    $U \sim \mathcal{U}(0,1)$
    **if** $U < \min\left(1, \overrightarrow{p}_{\theta,\phi}(\hat{\boldsymbol{x}}^{\sigma(d)}|\theta(\boldsymbol{x}^{\sigma(1:i)}), \phi(\hat{\boldsymbol{x}}^{\sigma(i+1:d-1)}))/\overleftrightarrow{p}_\theta(\hat{\boldsymbol{x}}^{\sigma(d)}|\theta(\boldsymbol{x}^{\sigma(1:i)}))\right)$ **then**
      Accept draft token $\boldsymbol{x}^{\sigma(d)} \leftarrow \hat{\boldsymbol{x}}^{\sigma(d)}$
    **else**
      Reject $\hat{\boldsymbol{x}}^{\sigma(d)}$ and resample, $\hat{\boldsymbol{x}}^{\sigma(d)} \sim \tilde{p}(\hat{\boldsymbol{x}}^{\sigma(d)})$ where
      $\tilde{p}(\hat{\boldsymbol{x}}^{\sigma(d)}) \propto \max\left(0, \overrightarrow{p}_{\theta,\phi}(\hat{\boldsymbol{x}}^{\sigma(d)}|\theta(\boldsymbol{x}^{\sigma(1:i)}), \phi(\hat{\boldsymbol{x}}^{\sigma(i+1:d-1)})) - \overleftrightarrow{p}_\theta(\hat{\boldsymbol{x}}^{\sigma(d)}|\theta(\boldsymbol{x}^{\sigma(1:i)}))\right)$
      Exit for loop and set $\boldsymbol{x}^{\sigma(d)} \leftarrow \hat{\boldsymbol{x}}^{\sigma(d)}$
    **end if**
  **end for**
  $i \leftarrow d$    Update the current number of tokens revealed
**end while**

---

### 3.3 SAMPLING PROCEDURE

We sample our hybrid architecture using Algorithm 2. The general procedure follows Algorithm 1, revealing tokens in update steps until the sequence is complete. In a given update step (with $i$ tokens revealed), the non-causal blocks first generate draft tokens for all unknown positions, $\hat{\boldsymbol{x}}^{\sigma(i+1:D)} \sim \overleftrightarrow{p}_\theta(\hat{\boldsymbol{x}}^{\sigma(i+1:D)}|\theta(\boldsymbol{x}^{\sigma(1:i)}))$. The causal blocks then give target probabilities for the drafted tokens $\{\overrightarrow{p}_{\theta,\phi}(\hat{\boldsymbol{x}}^{\sigma(d)}|\theta(\boldsymbol{x}^{\sigma(1:i)}), \phi(\hat{\boldsymbol{x}}^{\sigma(i+1:d-1)}))\}_{d=i+1}^D$. Finally, a speculative sampling inner loop accepts some of the drafted tokens. Appendix B details a more general windowed procedure.

Our procedure bears some similarity to self-speculative (Zhang et al., 2024; Elhoushi et al., 2024; Liu et al., 2024) and Medusa (Cai et al., 2024) style approaches where an initial subset of the network is used to parameterize a draft distribution which is then verified by the full transformer. However, using self-speculation for MDMs causes additional complexity because the target distribution given by Equation 6 changes depending on the number of revealed tokens. Consider the targets $\overrightarrow{p}_{\theta,\phi}(\boldsymbol{x}^{\sigma(i+1:D)}|\theta(\boldsymbol{x}^{\sigma(1:i)}), \phi)$ and $\overrightarrow{p}_{\theta,\phi}(\boldsymbol{x}^{\sigma(j+1:D)}|\theta(\boldsymbol{x}^{\sigma(1:j)}), \phi)$ for $j > i$. The factors for a fixed $d$ differ; i.e., importantly, we have $\overrightarrow{p}_{\theta,\phi}(\boldsymbol{x}^{\sigma(d)}|\theta(\boldsymbol{x}^{\sigma(1:i)}), \phi(\boldsymbol{x}^{\sigma(i+1:d-1)})) \neq \overrightarrow{p}_{\theta,\phi}(\boldsymbol{x}^{\sigma(d)}|\theta(\boldsymbol{x}^{\sigma(1:j)}), \phi(\boldsymbol{x}^{\sigma(j+1:d-1)}))$. This is because the non-causal blocks operate on different inputs (one sequence with $i$ revealed tokens, the other with $j$). So the hidden states passed to the causal blocks differ, causing the target distribution for position $\sigma(d)$ to change throughout the generation trajectory. This is not an issue for existing self-speculative models restricted to a single left-to-right ordering. All transformer blocks are causal in this case and never observe future tokens, meaning the target remains unchanged during generation.

### 3.4 THEORETICAL RESULTS

Let $p_{\theta,\phi}(\boldsymbol{x}^{\sigma(1:D)}|\sigma)$ be the probability of generating $\boldsymbol{x}^{\sigma(1:D)}$ in the ordering $\sigma$ under Algorithm 2. It is non-trivial to reason about $p_{\theta,\phi}(\boldsymbol{x}^{\sigma(1:D)}|\sigma)$ because the inner loop target distribution changes whenever a rejection occurs. Furthermore, it is the alignment between $\overleftrightarrow{p}_\theta$ and $\overrightarrow{p}_{\theta,\phi}$ that determines the rejection probability and so both $\overleftrightarrow{p}_\theta$ and $\overrightarrow{p}_{\theta,\phi}$ influence $p_{\theta,\phi}(\boldsymbol{x}^{\sigma(1:D)}|\sigma)$. Therefore, to calculate $p_{\theta,\phi}(\boldsymbol{x}^{\sigma(1:D)}|\sigma)$, it would seem that one would have to sum the contributions of $\overleftrightarrow{p}_\theta$ and $\overrightarrow{p}_{\theta,\phi}$ over all combinations of possible numbers of tokens revealed during each inner loop which would be intractable for $D >> 1$. Instead, we show in Proposition 3.1 that $p_{\theta,\phi}(\boldsymbol{x}^{\sigma(1:D)}|\sigma)$ can be tractably calculated with $D$ forward passes and $O(D^2)$ operations using a recursive decomposition. The proof is provided in Appendix C.1.

**Proposition 3.1.** *Consider the sampling scheme defined by Algorithm 2. For a given ordering $\sigma$, let $A^{\sigma(d)}$ denote the event that the token in position $\sigma(d)$ was accepted and let $R^{\sigma(d)}$ denote the event it was rejected and resampled. The distribution of samples output by Algorithm 2, $p_{\theta,\phi}(\boldsymbol{x}^{\sigma(1:D)}|\sigma)$, then admits the following decomposition (with $A^{\sigma(D+1:D)} := \emptyset$, $p_{\theta,\phi}(x^{\sigma(1:0)}, R^{\sigma(0)}) := 1$)*

$$p_{\theta,\phi}(\boldsymbol{x}^{\sigma(1:D)}|\sigma) = p_{\theta,\phi}(\boldsymbol{x}^{\sigma(1:D)}, A^{\sigma(1:D)})$$
$$+ \sum\nolimits_{d=1}^{D} p_{\theta,\phi}(\boldsymbol{x}^{\sigma(1:d)}, R^{\sigma(d)}) p_{\theta,\phi}(\boldsymbol{x}^{\sigma(d+1:D)}, A^{\sigma(d+1:D)}|\boldsymbol{x}^{\sigma(1:d)}, R^{\sigma(d)}),$$
$$(10)$$

*where* $p_{\theta,\phi}(\boldsymbol{x}^{\sigma(1:d)}, R^{\sigma(d)}) = \sum\nolimits_{k=1}^{d} p_{\theta,\phi}(\boldsymbol{x}^{\sigma(1:k-1)}, R^{\sigma(k-1)})$
$$\times p_{\theta,\phi}(\boldsymbol{x}^{\sigma(k:d)}, A^{\sigma(k:d-1)}, R^{\sigma(d)}|\boldsymbol{x}^{\sigma(1:k-1)}, R^{\sigma(k-1)}). \quad (11)$$

$p_{\theta,\phi}(\boldsymbol{x}^{\sigma(d+1:D)}, A^{\sigma(d+1:D)}|\boldsymbol{x}^{\sigma(1:d)}, R^{\sigma(d)})$ *and* $p_{\theta,\phi}(\boldsymbol{x}^{\sigma(k:d)}, A^{\sigma(k:d-1)}, R^{\sigma(d)}|\boldsymbol{x}^{\sigma(1:k-1)}, R^{\sigma(k-1)})$ *correspond to the likelihood associated with a single inner loop speculative sampling procedure. Their directly calculable forms in terms of $\overleftrightarrow{p}_\theta$ and $\overrightarrow{p}_{\theta,\phi}$ are given in Appendix C.1. Further, $p_{\theta,\phi}(\boldsymbol{x}^{\sigma(1:D)}|\sigma)$ can be computed with $D$ neural network forward passes and $O(D^2)$ operations.*

Proposition 3.1 is a dynamic programming recursion where the total likelihood $p_{\theta,\phi}(\boldsymbol{x}^{\sigma(1:D)}|\sigma)$ is split up using the simpler object $p_{\theta,\phi}(\boldsymbol{x}^{\sigma(1:d)}, R^{\sigma(d)})$ which can be written as a recursion on its previous values $p_{\theta,\phi}(\boldsymbol{x}^{\sigma(1:k-1)}, R^{\sigma(k-1)})$ for $k = 1$ to $d$. With a tractable expression for $p_{\theta,\phi}(\boldsymbol{x}^{\sigma(1:D)}|\sigma)$, we can now obtain an Evidence Lower BOund (ELBO) on $\log p_{\theta,\phi}(\boldsymbol{x})$,

$$\log p_{\theta,\phi}(\boldsymbol{x}) \geq \mathbb{E}_{p(\sigma)}\left[\log p_{\theta,\phi}(\boldsymbol{x}^{\sigma(1:D)}|\sigma)\right]. \quad (12)$$

We could use this ELBO as a training objective but favored instead the objective in (9) which is simpler and computationally much cheaper. Another quantity of theoretical interest is the number of inner loops of Algorithm 2 (each corresponding to one network forward pass) required to generate a sample, which defines the expected computational cost to generate a given datapoint. This requires knowing the total rejection count, $N^D$, which can be found again through a recursive decomposition of the model likelihood. In Appendix C.2 we derive an expression for $p_{\theta,\phi}(N^D|\boldsymbol{x}^{\sigma(1:D)}, \sigma)$ for a given generation and ordering.

### 3.5 SAMPLING IMPROVEMENTS

Algorithm 2 assumes each sampling inner loop corresponds to one forward pass of the non-causal blocks and a single verification pass through the remaining causal blocks. We can vary the sample quality-efficiency trade-off by running the speculative inner loop multiple times per single forward pass of the non-causal blocks. The non-causal forward pass first creates a draft $\overleftrightarrow{p}_\theta(\boldsymbol{x}^{\sigma(i+1:D)}|\theta(\boldsymbol{x}^{\sigma(1:i)}))$ distribution. Draft tokens are then sampled and we compute the target probabilities on these draft tokens using the causal layers. One speculative sampling loop is run to accept a number of these draft tokens finishing with a rejection and a resampling of a token. In the next speculative sampling inner loop we re-use the non-causal hidden states but recompute the target probabilities on the remaining draft tokens, noting the resampling operation will change the target probabilities for subsequent tokens compared to the first speculative sampling inner loop. This procedure is repeated for a user specified number of inner loops, the full algorithm is described in Appendix B. In our experiments, the vast majority of the network is non-causal, so this procedure greatly increases efficiency. To know when to stop generating tokens with the same draft $\overleftrightarrow{p}_\theta(\boldsymbol{x}^{\sigma(i+1:D)}|\theta(\boldsymbol{x}^{\sigma(1:i)}))$, we use windows (Leviathan et al., 2023; Chen et al., 2023) to set a maximum number of tokens we can reveal. We discuss window choices in Appendix D.

## 4 RELATED WORK

Several speculative sampling works have explored using a single network for both the draft model and target model (Zhang et al., 2024; Elhoushi et al., 2024; Liu et al., 2024), termed self-speculative decoding. These differ from our work by using causal draft and target layers in a strict left-to-right

setting. In the same setting, non-causal factorized predictions were explored by Cai et al. (2024) using a mostly causal stack with single-layer factorized prediction heads to draft tokens. We flip this: our network is mostly non-causal with a small final causal layer. Finally, a non-causal draft model and entirely separate causal target model was explored for audio generation in Ziv et al. (2024). We combine these two into a single architecture reducing deployment complexity.

Causal models for any-order AR training were explored by Yang et al. (2019), who introduced XLNet, a specialized two-stream architecture enforcing correct causal dependencies for different orderings. Further architectures used a standard single stream causal architecture and input ordering information through double positional encodings (Pannatier et al., 2024) or extra position tokens (Pang et al., 2025). Pannatier et al. (2024) describe a speculative sampling procedure but its validity is not established and the draft model is not described. An any-order causal transformer baseline reported by Hoogeboom et al. (2022) performed poorly in the MDMs context. We improve on the pure any-order causal model by letting it re-use computations from a powerful non-causal transformer.

Guo and Ermon (2025) perform self-speculative decoding within an any-subset AR model, which, given an arbitrarily located prompt, in-fills missing tokens in a left-to-right fashion. In contrast, our model can condition on an arbitrarily located prompt and in-fill using any ordering too, more akin to standard MDMs. They rely on the XLNet architecture which faces optimization difficulties for generation tasks with short prompts and large numbers of missing tokens requiring optimization tuning such as a masking schedule warm-up (Yang et al., 2019). Our architecture does not require any mask schedule warm-up and can operate in the fully masked regime with no prompt.

In the MDMs context, previous works have also aimed at speeding up sampling. Similar to our work, some aim to introduce dependencies between the newly sampled tokens via an energy-based model (Xu et al., 2025) or a latent variable mixture model (Hayakawa et al., 2025). We instead opt for speculative sampling to sample from a non-factorized target in one shot. Liu et al. (2025) use an AR model to derive a target, however, their predictive distribution must be sampled autoregressively for each denoising step. Deschenaux and Gulcehre (2025) use distillation to speed up sampling, however, their student model remains factorized, limiting potential speedups.

## 5 EXPERIMENTS

### 5.1 TEXT8

We first validate our method on the small-scale text8 dataset (100M chars, 27-token vocabulary). We trained a 150M parameter semi-causal transformer (11 non-causal blocks, 1 causal block) using the objective given in Equation 9. In Figure 2 we show the training losses split by the non-causal, $\log \overleftrightarrow{p}_\theta(\boldsymbol{x}^{\sigma(i+1:D)}|\theta(\boldsymbol{x}^{\sigma(1:i)}))$, and causal, $\log \overrightarrow{p}_{\theta,\phi}(\boldsymbol{x}^{\sigma(i+1:D)}|\theta(\boldsymbol{x}^{\sigma(1:i)}),\phi)$ components. The total loss is the sum of these two components. We see that in the initial stage of training the non-causal and causal loss exactly track each other, attributable to our use of a residual connection to the causal output in our architecture. After around $10^4$ steps, the causal block then learns to make use of the additional context available to it and dramatically reduces its training loss. This exemplifies the extra capacity a causal distribution has to fit the training data distribution due to it moving beyond the factorization assumption. With our speculative sampling procedure, we can efficiently produce samples from this more powerful causal distribution.

We now investigate our model's efficiency-sample quality trade-off. We compute the NFE during sampling by defining 1 NFE as a full 12-block forward pass. A standard speculative step (11 non-causal + 1 causal pass) counts as 1 NFE. Running the causal block multiple times increases this fractionally (e.g., 1 non-causal + 7 causal passes = 18/12=1.5 NFE). For MDMs, an update is 1 NFE unless no tokens change value in which case the update step could have been skipped and this is counted as 0 NFE. We used this best case analysis for MDM to provide a strong baseline.

To measure sample quality we calculate spelling accuracy by generating $D = 256$ characters and then computing the proportion of words within the sample that also appear in the training dataset; a word being defined as any number of lowercase characters between two whitespace characters. To create the accuracy-NFE tradeoff, we vary the window size and number of draft-verify steps (each requiring one forward pass of the causal block) undertaken per pass of the non-causal blocks. We

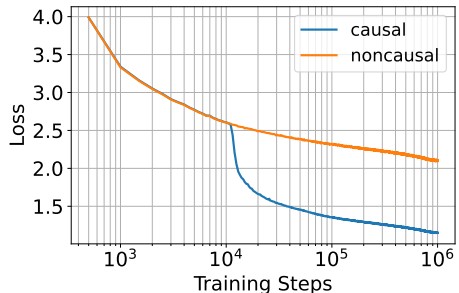
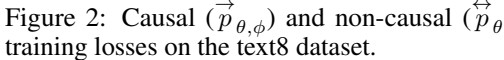
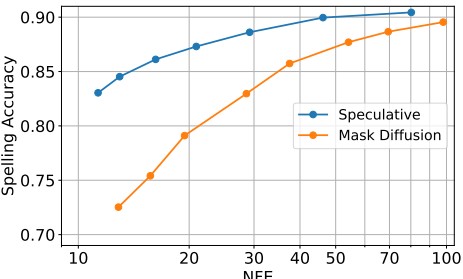

Figure 2: Causal ($\overrightarrow{p}_{\theta,\phi}$) and non-causal ($\overleftrightarrow{p}_{\theta}$) training losses on the text8 dataset.

Figure 3: Spelling accuracy versus number of function evaluations (NFE) on the Text8 dataset for our speculative approach and mask diffusion.

utilize a cosine shaped window (similar to the cosine discretization grid proposed in Shi et al. 2024) with varying time step parameter, the derivation of which is given in Appendix D.

We compare against MDM using the implementation from Shi et al. (2024) with a varying number of timesteps used to simulate the generative process. We plot our results in Figure 3. We see that our approach is able to achieve higher spelling accuracy at lower NFE versus standard MDMs. For example in the low NFE range, our approach is able to achieve greater than $2\times$ reduction in NFE.

## 5.2 OPENWEBTEXT

We demonstrate our method on OpenWebText using a GPT2-scale, 150M parameter, 12-layer transformer with RoPE positional encodings (Shi et al., 2024), setting the first 11 layers as non-causal and the final layer as causal. We use generative perplexity (measured by GPT2) as our metric, supplementing it with unigram token entropy to ensure sample diversity (as generative perplexity can be cheated by low-temperature sampling). Compared to an MDM baseline (Shi et al., 2024) with an identical architecture, our method (Table 1) achieves the same generative perplexity with half the NFE, while maintaining similar sample entropy for all NFE levels.

We also compare with Self-Distillation Through Time (SDTT) (Deschenaux and Gulcehre, 2025), where a student model is trained on a coarse time grid to match a teacher MDM sampled on a fine time grid. SDTT achieves very low GPT2 NLLs, far below the teacher MDM model using much larger NFE. Investigating the entropy of the produced samples reveals that SDTT samples are lower entropy than other methods. This mode-seeking is likely caused by truncation errors in the SDTT teacher model sampling (Zheng et al., 2025). In contrast, our approach reduces NFE while maintaining sample diversity equivalent to the baseline.

Table 1: GPT2 NLLs (in nats per dim) and unigram token entropy (nats). For each method a metric-NFE tradeoff curve is created by varying sampling parameters. Values at each NFE are read off by linearly interpolating between the two nearest points. For comparison we include results for mask diffusion and Self Distillation Through Time (SDTT) (Deschenaux and Gulcehre, 2025). We also include ablations, removing the residual output connection and a 10 non-causal 2 causal block model.

| Method | GPT2 NLL ($\downarrow$) | | | | Entropy | | | |
|---|---|---|---|---|---|---|---|---|
| | 32 NFE | 64 NFE | 128 NFE | 256 NFE | 32 NFE | 64 NFE | 128 NFE | 256 NFE |
| Masked Diffusion | 5.50 | 5.27 | 5.13 | 5.05 | 5.72 | 5.70 | 5.68 | 5.67 |
| Speculative (ours) | 5.28 | 5.12 | 5.05 | 5.02 | 5.72 | 5.70 | 5.68 | 5.66 |
| SDTT | 3.70 | 3.46 | 3.30 | 3.18 | 5.31 | 5.25 | 5.18 | 5.11 |
| No output residual | 5.36 | 5.16 | 5.10 | 5.05 | 5.74 | 5.70 | 5.68 | 5.67 |
| 10nc-2c layers | 5.34 | 5.16 | 5.06 | 5.03 | 5.71 | 5.68 | 5.67 | 5.66 |

Finally, we perform two architectural ablations. First, removing the output residual connection (see Figure 1) worsens the GPT2 NLL - NFE trade-off. We hypothesize that the residual connection both makes the target easier for the single causal layer to learn and aligns the draft/target distributions, boosting acceptance rates. Second, moving from an 11 non-causal blocks and 1 causal block to a 10 non-causal blocks and 2 causal blocks architecture also worsens the GPT2 NLL - NFE trade-off demonstrating that the best balance between the power of the draft distribution and power of the target distribution is achieved with only a single causal block applied to the non-causal stack.

### 5.3 PROTEIN SEQUENCE MODELLING

We now demonstrate our method on non-text data by modeling amino acid sequences from the UniRef50 dataset, following settings from Wang et al. (2024) (40M sequences, max length 1022). To demonstrate applicability to pretrained models, we take the 150M parameter, 30-layer ESM2-based model from Wang et al. (2024) (Lin et al., 2023), freeze its parameters and add a single causal block (carrying hidden states from the final non-causal block), training only this extra head.

We measure sample quality using the average pLDDT from ESMFold (Lin et al., 2023); a higher pLDDT indicates higher folding confidence, suggesting the sequence better follows the natural distribution. We compare this to the original Wang et al. (2024) non-causal model, sampled using the standard MDM algorithm (omitting resampling, temperature scaling, or confidence scoring for a consistent baseline). Our results (Figure 4) show a better sample quality-NFE trade-off than the standard algorithm, in particular achieving a $\sim 2\times$ speed-up for high pLDDT. This showcases the benefit of adding just a single causal block to a frozen pretrained network. We note that algorithms that also choose $\sigma$ can achieve higher pLDDT values, for example Peng et al. (2025), however we here focus on the standard MDM formulation with $\sigma$ selected uniformly at random.

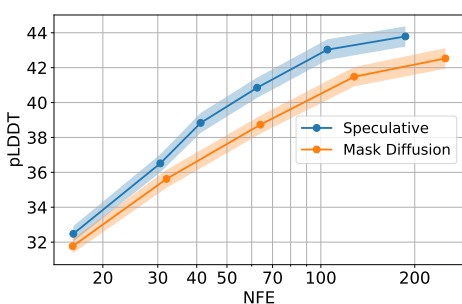

Figure 4: pLDDTs versus NFE for mask diffusion and our speculative method. The mean pLDDT is computed over 512 samples with the standard error of the mean represented by the shading.

## 6 DISCUSSION

We presented self-speculative masked diffusions, a novel architecture and sampling scheme for MDMs that reduces the number of network forward passes required for a given sample quality compared to standard MDM approaches. On a variety of datasets, from text to protein sequences, our method consistently improves computational efficiency, reducing the number of network evaluations by up to $2\times$ compared to MDMs while adding only a negligible computational overhead (a 0.98% increase in FLOPs in standard settings). Further work could explore the natural combination of our method with compute-intensive inference-scaling techniques, such as re-masking corrector steps, in order to reach a new level of model reasoning capability for a fixed compute budget.

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

# Appendix

This appendix is organized as follows. In Section A, we present an illustration of the different attention mechanisms considered in this paper. Section B details the complete speculative sampling algorithm, including the windowing technique used in our experiments. Section C then provides proofs for all of our propositions, deriving the likelihood of our model when sampled using Algorithm 2 and deriving the distribution over the number of rejection steps within Algorithm 2. In Section D we discuss choices for the window function used during sampling including the derivation of a cosine style window. In Section E we provide a FLOP analysis of the additional overhead of our self-speculative architecture and in Section F we discuss the influence of hyperparameters on our method. Finally, details for all of our experiments are provided in Section G.

## A    ILLUSTRATION OF ATTENTION MECHANISMS

In order to give an intuitive illustration of the different mechanisms at play in masked diffusion, classical AR models and AR models with random ordering we present the attention matrix (with its mask) in Figure 5. We consider the text sequence from Figure 1, *"Speculation is like hazarding a guess"*.

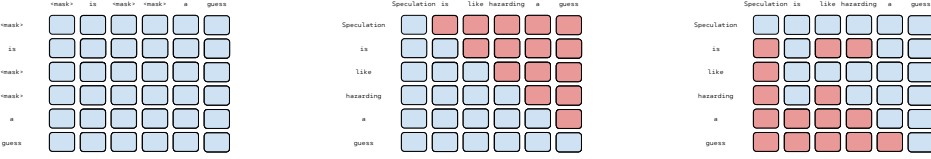

Figure 5: From left to right: The attention mechanism based on the example described in Figure 1. From left to right: any-to-any attention masks, standard left-to-right attention masks for AR models and a causal attention mask applied to the ordering from Figure 1. Rows correspond to the token making the query (attending). Columns correspond to the token providing the key (being attended to).

The first case on the left of Figure 5 is the non-causal any-to-any attention mask for standard MDMs. Each token position is able to attend to the values in all other positions represented by blue squares. The correct dependencies in the $\overleftrightarrow{p}_\theta(x^{\sigma(i+1:D)}|\theta(x^{\sigma(1:i)}))$ distribution are induced through masking the tokens that are not present in the conditioning information $x^{\sigma(1:i)}$.

The second case in the middle of Figure 5 is a causal left-to-right attention mask for classical AR models that only consider a linear left-to-right ordering, $\sigma = \{1, 2, \ldots, D\}$. Each row corresponds to a token making a query against the values along the top. For example, the token *"Speculation"* can attend only to itself whereas the token *"like"* can attend to all of *"Speculation"*, *"is"*, and *"like"*. Since a token can attend to its own value, this track of the transformer predicts the value in the next position in the ordering, e.g. the track corresponding to the token *"Speculation"* predicts the token *"is"* and the track corresponding to the token *"like"* predicts the token *"hazarding"*. We note this is different to the non-causal attention mask case on the left of Figure 5 where each track predicts the token in its own position.

Finally, we represent the case of a causal attention mask applied to an arbitrary ordering on the right of Figure 5. We consider the ordering of Figure 1:

$$\sigma(1) = 6 \quad \text{guess}$$
$$\sigma(2) = 5 \quad \text{a}$$
$$\sigma(3) = 2 \quad \text{is}$$
$$\sigma(4) = 4 \quad \text{hazarding}$$
$$\sigma(5) = 3 \quad \text{like}$$
$$\sigma(6) = 1 \quad \text{Speculation}$$

Each token can only attend to itself and those tokens that precede it in the ordering. For example token *"a"* which is 2-nd in the $\sigma$ ordering can attend to tokens *"a"* and *"guess"* whereas token *"like"* which is 5-th in the $\sigma$ ordering can attend to all of *"guess"*, *"a"*, *"is"*, *"hazarding"* and *"like"*. As for the classical AR case, since a token can attend to itself, we have that track of the transformer predict the token that comes next in the $\sigma$ ordering. For example, the track corresponding to token *"a"* predicts *"is"* and the track corresponding to token *"like"* predicts *"Speculation"*. Since the attention mask imparts the required conditional dependencies, no explicit mask tokens are required in the input to this style of transformer block. We note that in practice, this style of attention is achieved with a standard left-to-right attention matrix but operating on the permuted sequence which is the scheme depicted in Figure 1.

## B  FULL SAMPLING ALGORITHM

In this section we present the full sampling algorithm with additional hyperparameters as detailed in Section 3.5. We require a window function $W(i)$ that takes in the current number of revealed tokens $i$ and outputs the maximum number of tokens allowed to be revealed for this current pass of the non-causal blocks. We discuss choices for $W(i)$ in Appendix D. We also require a constant $N$ that is the number of draft-verify steps to perform per forward pass of the non-causal blocks. The number of causal forward passes per non-causal forward pass is then equal to $N$.

---

**Algorithm 3** Self-Speculative Sampling Masked Diffusion Sampling - Full Procedure

---

$i \leftarrow 0$    Number of tokens currently revealed
$\sigma \sim p(\sigma)$    Draw the generation ordering
**while** $i < D$ **do**
    $\hat{\boldsymbol{x}}^{\sigma(i+1:\min(i+W(i),D))} \sim \overleftrightarrow{p}_\theta(\hat{\boldsymbol{x}}^{\sigma(i+1:\min(i+W(i),D))}|\theta(\boldsymbol{x}^{\sigma(1:i)}))$    Non-causal forward pass
    $j \leftarrow i$    Revealed tokens counter for internal loop
    **for** $n = 1$ to $N$ **do**
        Compute   $\{\overrightarrow{p}_{\theta,\phi}(\hat{\boldsymbol{x}}^{\sigma(d)}|\theta(\boldsymbol{x}^{\sigma(1:i)}),\phi(\hat{\boldsymbol{x}}^{\sigma(i+1:d-1)}))\}_{d=j+1}^{\min(i+W(i),D)}$    Causal forward pass
        **for** $d = j+1$ to $\min(i+W(i),D)$ **do**
          $U \sim \mathcal{U}(0,1)$
          **if** $U < \min\left(1, \overrightarrow{p}_{\theta,\phi}(\hat{\boldsymbol{x}}^{\sigma(d)}|\theta(\boldsymbol{x}^{\sigma(1:i)}),\phi(\hat{\boldsymbol{x}}^{\sigma(i+1:d-1)}))/\overleftrightarrow{p}_\theta(\hat{\boldsymbol{x}}^{\sigma(d)}|\theta(\boldsymbol{x}^{\sigma(1:i)}))\right)$ **then**
            Accept draft token $\boldsymbol{x}^{\sigma(d)} \leftarrow \hat{\boldsymbol{x}}^{\sigma(d)}$
          **else**
            Reject $\hat{\boldsymbol{x}}^{\sigma(d)}$ and resample $\hat{\boldsymbol{x}}^{\sigma(d)} \sim \tilde{p}(\hat{\boldsymbol{x}}^{\sigma(d)})$ where
              $\tilde{p}(\hat{\boldsymbol{x}}^{\sigma(d)}) \propto \max\left(0, \overrightarrow{p}_{\theta,\phi}(\hat{\boldsymbol{x}}^{\sigma(d)}|\theta(\boldsymbol{x}^{\sigma(1:i)}),\phi(\hat{\boldsymbol{x}}^{\sigma(i+1:d-1)})) - \overleftrightarrow{p}_\theta(\hat{\boldsymbol{x}}^{\sigma(d)}|\theta(\boldsymbol{x}^{\sigma(1:i)}))\right)$
            Set $\boldsymbol{x}^{\sigma(d)} \leftarrow \hat{\boldsymbol{x}}^{\sigma(d)}$
            Exit "$d = j+1$ to $\min(i+W(i),D)$" for loop
          **end if**
        **end for**
        $j \leftarrow d$    Update internal counter of current number of tokens revealed
    **end for**
    $i \leftarrow j$    Update outer counter of current number of tokens revealed
**end while**

---

Algorithm 3 consists of an outer loop each corresponding to one forward pass of the non-causal layers and an inner speculative sampling loop with $N$ iterations. When the current number of

revealed tokens is $i$, the outer loop starts with a forward pass of the non-causal blocks to obtain the draft distribution for the currently masked tokens up to a maximum length specified by the window function, $\overleftrightarrow{p}_\theta(\hat{\boldsymbol{x}}^{\sigma(i+1:\min(i+W(i),D))}|\theta(\boldsymbol{x}^{\sigma(1:i)}))$. The algorithm then begins the $N$ inner speculative sampling loops.

The inner speculative sampling loop begins with a forward pass of the causal layers re-using the non-causal hidden states that were computed on $\boldsymbol{x}^{\sigma(1:i)}$. In addition, the causal layers get access to the values of the drafted tokens. The causal probabilities are then of the form

$$\{\overrightarrow{p}_{\theta,\phi}(\hat{\boldsymbol{x}}^{\sigma(d)}|\theta(\boldsymbol{x}^{\sigma(1:i)}),\phi(\hat{\boldsymbol{x}}^{\sigma(i+1:d-1)}))\}_{d=j+1}^{\min(i+W(i),D)} \tag{13}$$

With these target probabilities, a speculative sampling procedure can then be run which will accept some number of the draft tokens. Once a rejection occurs, that draft token is resampled from the resample probability distribution

$$\tilde{p}(\hat{\boldsymbol{x}}^{\sigma(d)}) \propto \max\left(0, \overrightarrow{p}_{\theta,\phi}(\hat{\boldsymbol{x}}^{\sigma(d)}|\theta(\boldsymbol{x}^{\sigma(1:i)}),\phi(\hat{\boldsymbol{x}}^{\sigma(i+1:d-1)})) - \overleftrightarrow{p}_\theta(\hat{\boldsymbol{x}}^{\sigma(d)}|\theta(\boldsymbol{x}^{\sigma(1:i)}))\right). \tag{14}$$

The token $\hat{\boldsymbol{x}}^{\sigma(d)}$ is set to this new value that has been sampled from $\tilde{p}(\hat{\boldsymbol{x}}^{\sigma(d)})$. The next iteration of speculative sampling can then begin. We perform another forward pass of the causal layers using the same non-causal hidden states as before. However, due to the resampling, one of the draft tokens has changed value and hence the causal probabilities are different than the previous speculative sampling loop. We repeat the speculative sampling procedure with these new causal target probabilities, noting the non-causal draft probabilities remain unchanged however.

These speculative sampling loops repeat until either $N$ loops have been completed or $\min(i+W(i),D)$ tokens have been revealed. In the next outer iteration, the non-causal layers are applied to the currently revealed tokens including those that were just revealed in the prior $N$ speculative sampling inner loops and the algorithm repeats.

## C  PROOFS

### C.1  PROOF OF PROPOSITION 3.1

We aim to find the likelihood assigned to a given datapoint $\boldsymbol{x}^{1:D}$ under the generative model defined by models $\overrightarrow{p}_{\theta,\phi}$, $\overleftrightarrow{p}_\theta$, sampling Algorithm 2 and ordering $\sigma$. Let us first start by assuming that we have a fixed linear ordering $\sigma = \{1,\ldots,D\}$ for simplicity. We let $A_d$ denote the event that for the $d$-th dimension, we accepted the draft token in the speculative sampling procedure. Let $R_d$ correspondingly denote the event that in the $d$-th dimension we rejected the draft token.

We note that if we have event $A_d$, then for dimension $d+1$ we will be continuing in the same speculative sampling loop as for dimension $d$. If we have event $R_d$, on the other hand, we will exit the speculative sampling loop at dimension $d$ and then enter into the next iteration of the outer loop in Algorithm 2. The significance of this outer loop iteration change is that the draft distribution changes from $\overleftrightarrow{p}_\theta(\boldsymbol{x}^{i+1:D}|\theta(\boldsymbol{x}^{1:i}))$ to $\overleftrightarrow{p}_\theta(\boldsymbol{x}^{d+1:D}|\theta(\boldsymbol{x}^{1:d}))$, with $d > i$ meaning additional context is now revealed to the non-causal blocks. In other words, once we have a rejection, then the draft distribution will update with all the new tokens that were produced in the previous speculative sampling run. Therefore, in order to know the correct distribution that is being used as the draft, we need to know when our last rejection was to know what conditioning information is being input into the non-causal blocks.

In order to make headway into this problem, we utilize a recursive decomposition approach exploiting the fact that the likelihood of $\boldsymbol{x}^d$ and $A^d$ is independent of all accept/reject decisions prior to the most recent reject decision. This is because $\overleftrightarrow{p}_\theta$ and $\overrightarrow{p}_{\theta,\phi}$ depend only on the currently revealed tokens $\boldsymbol{x}^{1:i}$ and not the exact path taken to generate those tokens.

In our proof, we will let $Y^d$ be the result of the accept/reject step for dimension $d$ i.e. $Y^d \in \{A^d, R^d\}$. As a shorthand, we will sometimes simply write $A^d$ to mean $Y^d = A^d$ and $R^d$ to mean $Y^d = R^d$.

We start by introducing some marginalization over the accept/reject decisions,

$$p_{\theta,\phi}(\boldsymbol{x}^{1:D}) \tag{15}$$

$$= \sum_{Y^{1:D}} p_{\theta,\phi}(\boldsymbol{x}^{1:D}, Y^{1:D}) \tag{16}$$

$$= \sum_{Y^{1:D-1}} p_{\theta,\phi}(\boldsymbol{x}^{1:D}, Y^{1:D-1}, A^D) + \sum_{Y^{1:D-1}} p_{\theta,\phi}(\boldsymbol{x}^{1:D}, Y^{1:D-1}, R^D) \tag{17}$$

$$= \sum_{Y^{1:D-1}} p_{\theta,\phi}(\boldsymbol{x}^{1:D}, Y^{1:D-1}, A^D) + p_{\theta,\phi}(\boldsymbol{x}^{1:D}, R^D) \tag{18}$$

$$= \sum_{Y^{1:D-2}} p_{\theta,\phi}(\boldsymbol{x}^{1:D}, Y^{1:D-2}, A^{D-1}, A^D) + \sum_{Y^{1:D-2}} p_{\theta,\phi}(\boldsymbol{x}^{1:D}, Y^{1:D-2}, R^{D-1}, A^D) \tag{19}$$

$$+ p_{\theta,\phi}(\boldsymbol{x}^{1:D}, R^D) \tag{20}$$

$$= \sum_{Y^{1:D-2}} p_{\theta,\phi}(\boldsymbol{x}^{1:D}, Y^{1:D-2}, A^{D-1}, A^D) + p_{\theta,\phi}(\boldsymbol{x}^{1:D}, R^{D-1}, A^D) + p_{\theta,\phi}(\boldsymbol{x}^{1:D}, R^D) \tag{21}$$

$$= \quad \vdots \tag{22}$$

$$= p_{\theta,\phi}(\boldsymbol{x}^{1:D}, A^{1:D}) + \sum_{d=1}^{D} p_{\theta,\phi}(\boldsymbol{x}^{1:D}, R^d, A^{d+1:D}) \tag{23}$$

$$= p_{\theta,\phi}(\boldsymbol{x}^{1:D}, A^{1:D}) + \sum_{d=1}^{D} p_{\theta,\phi}(\boldsymbol{x}^{1:d}, R^d) p_{\theta,\phi}(\boldsymbol{x}^{d+1:D}, A^{d+1:D}|\boldsymbol{x}^{1:d}, R^d) \tag{24}$$

The distribution $p_{\theta,\phi}(\boldsymbol{x}^{d+1}, A^{d+1:D}|\boldsymbol{x}^{1:d}, R^d)$ is easy to calculate because it is the probability of generating the given $\boldsymbol{x}^{d+1:D}$ and all of them being accepted when the last rejection was $R^d$. Since the last rejection is the same for all the $d+1:D$ dimensions, the conditioning information for the draft model is not changing since all the dimensions are being generated in the same inner speculative sampling procedure.

Specifically, we have

$$p_{\theta,\phi}(\boldsymbol{x}^{d+1:D}, A^{d+1:D}|\boldsymbol{x}^{1:d}, R^d) = \prod_{k=d+1}^{D} p_{\theta,\phi}(A^k, \boldsymbol{x}^k|x^{1:k-1}, R^d, A^{d+1:k-1}) \tag{25}$$

We now introduce Lemma C.1 that allows us to compute the joint probability of having an acceptance in position $k$ and the output token having value $\boldsymbol{x}^k$.

**Lemma C.1.** *For a single speculative sampling accept/reject step with target distribution $\overrightarrow{p}$ and draft distribution $\overleftrightarrow{p}$, we have that the joint probability over the output token and that an acceptance occurs is*

$$p(\boldsymbol{x}, A) = \min(\overleftrightarrow{p}(\boldsymbol{x}), \overrightarrow{p}(\boldsymbol{x})), \tag{26}$$

*while the joint probability over the output token and that a rejection occurs is*

$$p(\boldsymbol{x}, R) = \max(0, \overrightarrow{p}(\boldsymbol{x}) - \overleftrightarrow{p}(\boldsymbol{x})). \tag{27}$$

*Proof.* A single speculative sampling step consists of 1) Sampling a draft token $\hat{\boldsymbol{x}} \sim \overleftrightarrow{p}(\hat{\boldsymbol{x}})$, 2) Accepting $\hat{\boldsymbol{x}}$ with probability $\min\left(1, \frac{\overrightarrow{p}(\hat{\boldsymbol{x}})}{\overleftrightarrow{p}(\hat{\boldsymbol{x}})}\right)$, 3) If rejected, resampling a new token $\tilde{\boldsymbol{x}}$ from the residual distribution $p_{\text{res}}(\tilde{\boldsymbol{x}}) \propto \max(0, \overrightarrow{p}(\tilde{\boldsymbol{x}}) - \overleftrightarrow{p}(\tilde{\boldsymbol{x}}))$. For an accepted token $\boldsymbol{x}$, the joint probability $p(\boldsymbol{x}, A)$ is simply the probability of drafting $\boldsymbol{x}$ and then accepting it:

$$p(\boldsymbol{x}, A) = \overleftrightarrow{p}(\boldsymbol{x}) \min\left(1, \frac{\overrightarrow{p}(\boldsymbol{x})}{\overleftrightarrow{p}(\boldsymbol{x})}\right) = \min(\overleftrightarrow{p}(\boldsymbol{x}), \overrightarrow{p}(\boldsymbol{x})). \tag{28}$$

For a rejected and resampled token $\boldsymbol{x}$, it must be drawn from $p_{\text{res}}(\boldsymbol{x})$ after some draft token is rejected. The total probability of rejection, $P(R)$, is the expected rejection rate over all possible draft tokens:

$$P(R) = \sum_{\hat{\boldsymbol{x}}} \overset{\leftrightarrow}{p}(\hat{\boldsymbol{x}}) \left( 1 - \min \left( 1, \frac{\overrightarrow{p}(\hat{\boldsymbol{x}})}{\overset{\leftrightarrow}{p}(\hat{\boldsymbol{x}})} \right) \right) \tag{29}$$

$$= \sum_{\hat{\boldsymbol{x}}} \left( \overset{\leftrightarrow}{p}(\hat{\boldsymbol{x}}) - \min(\overset{\leftrightarrow}{p}(\hat{\boldsymbol{x}}), \overrightarrow{p}(\hat{\boldsymbol{x}})) \right) \tag{30}$$

$$= \sum_{\hat{\boldsymbol{x}}} \max(0, \overset{\leftrightarrow}{p}(\hat{\boldsymbol{x}}) - \overrightarrow{p}(\hat{\boldsymbol{x}})) \tag{31}$$

$$= \sum_{\hat{\boldsymbol{x}}} \max(0, \overrightarrow{p}(\hat{\boldsymbol{x}}) - \overset{\leftrightarrow}{p}(\hat{\boldsymbol{x}})) \quad \text{as} \quad \sum_{\hat{\boldsymbol{x}}} (\overrightarrow{p}(\hat{\boldsymbol{x}}) - \overset{\leftrightarrow}{p}(\hat{\boldsymbol{x}})) = 0. \tag{32}$$

Finally, given $p(\boldsymbol{x}, R) = P(R) \cdot p_{\text{res}}(\boldsymbol{x})$, we obtain

$$p(\boldsymbol{x}, R) = P(R) \cdot \frac{\max(0, \overrightarrow{p}(\boldsymbol{x}) - \overset{\leftrightarrow}{p}(\boldsymbol{x}))}{P(R)} = \max(0, \overrightarrow{p}(\boldsymbol{x}) - \overset{\leftrightarrow}{p}(\boldsymbol{x})). \tag{33}$$

This completes the proof. $\qquad \square$

We can now continue with our proof of Proposition 3.1, namely by using Lemma C.1, we have

$$\prod_{k=d+1}^{D} p_{\theta,\phi}(A^k, \boldsymbol{x}^k | x^{1:k-1}, R^d, A^{d+1:k-1}) \tag{34}$$

$$= \prod_{k=d+1}^{D} \min \left( \overset{\leftrightarrow}{p}_{\theta}(\boldsymbol{x}^k | \theta(\boldsymbol{x}^{1:d})), \overrightarrow{p}_{\theta,\phi}(\boldsymbol{x}^k | \theta(\boldsymbol{x}^{1:d}), \phi(\boldsymbol{x}^{d+1:k-1})) \right). \tag{35}$$

Therefore, we can calculate $p_{\theta,\phi}(\boldsymbol{x}^{d+1:D}, A^{d+1:D} | \boldsymbol{x}^{1:d}, R^d)$ in $O(D)$ operations.

We now deal with the term $p_{\theta,\phi}(\boldsymbol{x}^{1:d}, R^d)$. We perform a recursive decomposition,

$$p_{\theta,\phi}(\boldsymbol{x}^{1:d}, R^d) \tag{36}$$

$$= \sum_{Y^{1:d-1}} p_{\theta,\phi}(\boldsymbol{x}^{1:d}, Y^{1:d-1}, R^d) \tag{37}$$

$$= \sum_{Y^{1:d-2}} p_{\theta,\phi}(\boldsymbol{x}^{1:d}, Y^{1:d-2}, A^{d-1}, R^d) + \sum_{Y^{1:d-2}} p_{\theta,\phi}(\boldsymbol{x}^{1:d}, Y^{1:d-2}, R^{d-1}, R^d) \tag{38}$$

$$= \sum_{Y^{1:d-2}} p_{\theta,\phi}(\boldsymbol{x}^{1:d}, Y^{1:d-2}, A^{d-1}, R^d) + p_{\theta,\phi}(\boldsymbol{x}^{1:d}, R^{d-1}, R^d) \tag{39}$$

$$= \sum_{Y^{1:d-2}} p_{\theta,\phi}(\boldsymbol{x}^{1:d}, Y^{1:d-2}, A^{d-1}, R^d) + p_{\theta,\phi}(\boldsymbol{x}^{1:d-1}, R^{d-1}) p_{\theta,\phi}(\boldsymbol{x}^d, R^d | \boldsymbol{x}^{1:d-1}, R^{d-1}) \tag{40}$$

$$= \sum_{Y^{1:d-3}} p_{\theta,\phi}(\boldsymbol{x}^{1:d}, Y^{1:d-3}, A^{d-2}, A^{d-1}, R^d) + \sum_{Y^{1:d-3}} p_{\theta,\phi}(\boldsymbol{x}^{1:d}, Y^{1:d-3}, R^{d-2}, A^{d-1}, R^d) \tag{41}$$

$$+ p_{\theta,\phi}(\boldsymbol{x}^{1:d-1}, R^{d-1}) p_{\theta,\phi}(\boldsymbol{x}^d, R^d | \boldsymbol{x}^{1:d-1}, R^{d-1}) \tag{42}$$

$$= \sum_{Y^{1:d-3}} p_{\theta,\phi}(\boldsymbol{x}^{1:d}, Y^{1:d-3}, A^{d-2}, A^{d-1}, R^d) \tag{43}$$

$$+ p_{\theta,\phi}(\boldsymbol{x}^{1:d-2}, R^{d-2}) p_{\theta,\phi}(\boldsymbol{x}^{d-1:d}, A^{d-1}, R^d | \boldsymbol{x}^{1:d-2}, R^{d-2}) \tag{44}$$

$$+ p_{\theta,\phi}(\boldsymbol{x}^{1:d-1}, R^{d-1}) p_{\theta,\phi}(\boldsymbol{x}^d, R^d | \boldsymbol{x}^{1:d-1}, R^{d-1}) \tag{45}$$

$$= \quad \vdots \tag{46}$$

$$= \sum_{k=1}^{d} p_{\theta,\phi}(\boldsymbol{x}^{1:k-1}, R^{k-1}) p_{\theta,\phi}(\boldsymbol{x}^{k:d}, A^{k:d-1}, R^d | \boldsymbol{x}^{1:k-1}, R^{k-1}), \tag{47}$$

where $p_{\theta,\phi}(\boldsymbol{x}^{1:0}, R^0) = 1$ and $p_{\theta,\phi}(\boldsymbol{x}^{d:d}, A^{d:d-1}, R^d|\boldsymbol{x}^{1:d-1}, R^{d-1}) = p_{\theta,\phi}(\boldsymbol{x}^d, R^d|\boldsymbol{x}^{1:d-1}, R^{d-1})$ by definition.

We can efficiently compute all the required values of $p_{\theta,\phi}(\boldsymbol{x}^{k:d}, A^{k:d-1}, R^d|\boldsymbol{x}^{1:k-1}, R^{k-1})$ for all $d \geq k$. We first split each term as

$$p_{\theta,\phi}(\boldsymbol{x}^{k:d}, A^{k:d-1}, R^d|\boldsymbol{x}^{1:k-1}, R^{k-1}) \tag{48}$$

$$=p_{\theta,\phi}(\boldsymbol{x}^{k:d-1}, A^{k:d-1}|\boldsymbol{x}^{1:k-1}, R^{k-1})p_{\theta,\phi}(\boldsymbol{x}^d, R^d|\boldsymbol{x}^{1:d-1}, A^{k:d-1}, R^{k-1}). \tag{49}$$

By Lemma C.1, we have that

$$p_{\theta,\phi}(\boldsymbol{x}^{k:d-1}, A^{k:d-1}|\boldsymbol{x}^{1:k-1}, R^{k-1}) \tag{50}$$

$$= \prod_{l=k}^{d-1} p_{\theta,\phi}(\boldsymbol{x}^l, A^l|\boldsymbol{x}^{1:l-1}, A^{k:l-1}, R^{k-1}) \tag{51}$$

$$= \prod_{l=k}^{d-1} \min\left(\overleftrightarrow{p}_\theta(\boldsymbol{x}^l|\theta(\boldsymbol{x}^{1:k-1})), \overrightarrow{p}_{\theta,\phi}(\boldsymbol{x}^l|\theta(\boldsymbol{x}^{1:k-1}), \phi(\boldsymbol{x}^{k:l-1}))\right). \tag{52}$$

Therefore, we can just keep extending this product and obtain all of

$$\left\{p_{\theta,\phi}(\boldsymbol{x}^{k:d-1}, A^{k:d-1}|\boldsymbol{x}^{1:k-1}, R^{k-1})\right\}_{d=k+1}^{D} \tag{53}$$

in just $O(D)$ operations.

To obtain the required $p_{\theta,\phi}(\boldsymbol{x}^{k:d}, A^{k:d-1}, R^d|\boldsymbol{x}^{1:k-1}, R^{k-1})$ distributions in Equation 49 , we need to compute the additional rejection factor, for which we again make use of Lemma C.1

$$p_{\theta,\phi}(\boldsymbol{x}^d, R^d|\boldsymbol{x}^{1:d-1}, A^{k:d-1}, R^{k-1}) \tag{54}$$

$$= \max\left(0, \overrightarrow{p}_{\theta,\phi}(\boldsymbol{x}^d|\theta(\boldsymbol{x}^{1:k-1}), \phi(\boldsymbol{x}^{k:d-1})) - \overleftrightarrow{p}_\theta(\boldsymbol{x}^d|\theta(\boldsymbol{x}^{1:k-1}))\right). \tag{55}$$

We can therefore obtain

$$\left\{p_{\theta,\phi}(\boldsymbol{x}^{k:d}, A^{k:d-1}, R^d|\boldsymbol{x}^{1:k-1}, R^{k-1})\right\}_{d=k}^{D} \tag{56}$$

in $O(D)$ operations. Allowing $k$ to vary as well, we can obtain

$$\left\{p_{\theta,\phi}(\boldsymbol{x}^{k:d}, A^{k:d-1}, R^d|\boldsymbol{x}^{1:k-1}, R^{k-1})\right\}_{k \leq d} \tag{57}$$

in $O(D^2)$ operations.

Once, we have all of these values pre-computed we can use our formula for $p_{\theta,\phi}(\boldsymbol{x}^{1:d}, R^d)$ in Equation 47 to compute all of

$$\left\{p_{\theta,\phi}(\boldsymbol{x}^{1:d}, R^d)\right\}_{d=1}^{D} \tag{58}$$

in $O(D^2)$ operations. The total cost so far is then $O(D^2 + D^2) = O(D^2)$.

The final point of order is obtaining $p_{\theta,\phi}(\boldsymbol{x}^{1:D}, A^{1:D})$. By Lemma C.1 we have

$$p_{\theta,\phi}(\boldsymbol{x}^{1:D}, A^{1:D}) = \prod_{d=1}^{D} p_{\theta,\phi}(\boldsymbol{x}^d, A^d|\boldsymbol{x}^{1:d-1}, A^{1:d-1}) \tag{59}$$

$$= \prod_{d=1}^{D} \min\left(\overleftrightarrow{p}_\theta(\boldsymbol{x}^d|\theta(\emptyset)), \overrightarrow{p}_{\theta,\phi}(\boldsymbol{x}^d|\theta(\emptyset), \phi(\boldsymbol{x}^{1:d-1}))\right). \tag{60}$$

We now have all the required quantities to compute the full likelihood as

$$p_{\theta,\phi}(\boldsymbol{x}^{1:D}) = p_{\theta,\phi}(\boldsymbol{x}^{1:D}, A^{1:D}) + \sum_{d=1}^{D} p_{\theta,\phi}(\boldsymbol{x}^{1:d}, R^d)p_{\theta,\phi}(\boldsymbol{x}^{d+1:D}, A^{d+1:D}|\boldsymbol{x}^{1:d}, R^d). \quad (61)$$

Computing all the $p_{\theta,\phi}(\boldsymbol{x}^{1:d}, R^d)$ quantities cost $O(D^2)$ operations. Computing $p_{\theta,\phi}(\boldsymbol{x}^{d+1:D}, A^{d+1:D}|\boldsymbol{x}^{1:d}, R^d)$ cost $O(D)$ each so computing all these required factors is another $O(D^2)$ cost. Computing $p_{\theta,\phi}(\boldsymbol{x}^{1:D}, A^{1:D})$ is just $O(D)$. Therefore, the total cost of getting the full likelihood is $O(D^2)$.

An important note is to count how many forward passes of the neural networks are required. The required distributions are

$$\overleftrightarrow{p}_{\theta}(\boldsymbol{x}^d|\theta(\boldsymbol{x}^{1:k})) \quad \forall d > k \quad (62)$$

$$\overrightarrow{p}_{\theta,\phi}(\boldsymbol{x}^d|\theta(\boldsymbol{x}^{1:k}), \phi(\boldsymbol{x}^{k+1:d-1})) \quad \forall d > k \quad (63)$$

which in total is only $O(D)$ forward passes. So although we need to perform $O(D^2)$ total operations, there are only $O(D)$ expensive neural network operations required.

Now we remove the assumption that the ordering is fixed when calculating the likelihood. Let us assume some distribution over orderings, $\sigma \sim p(\sigma)$ that we use to calculate our likelihood e.g. uniform. A bound on the log-likelihood is therefore

$$\log p_{\theta,\phi}(\boldsymbol{x}^{1:D}) \geq \mathbb{E}_{p(\sigma)}\left[\log p_{\theta,\phi}(\boldsymbol{x}^{1:D}|\sigma)\right] \quad (64)$$

For a given $\sigma$, we can calculate $\log p_{\theta,\phi}(\boldsymbol{x}^{1:D}|\sigma)$ as above, just assuming we generate in the ordering given by $\sigma$. Therefore, we can obtain an estimate of a lower bound on the log-likelihood by sampling $\sigma$ and calculating $\log p_{\theta,\phi}(\boldsymbol{x}^{1:D}|\sigma)$. To compute the log of sums of probabilities, we use LogSumExp for numerical stability.

C.2   NUMBER OF REJECTIONS

Here we derive an expression for the distribution over the number of rejections during the speculative sampling procedure given in Algorithm 2. The number of forward passes required to generate $\boldsymbol{x}^{1:D}$ is then one more than the number of rejections. We now formally state and prove our result.

**Proposition C.2.** *Consider the sampling scheme defined by Algorithm 2. Let $N^d$ be the number of rejections occurring in the algorithm up until dimension $d$ in the ordering. The posterior of $N^d$ given a datapoint and specific ordering is*

$$p_{\theta,\phi}(N^D|\boldsymbol{x}^{\sigma(1:D)},\sigma) = \frac{p_{\theta,\phi}(\boldsymbol{x}^{1:D},N^D|\sigma)}{p_{\theta,\phi}(\boldsymbol{x}^{1:D}|\sigma)} \tag{65}$$

*where*

$$p_{\theta,\phi}(\boldsymbol{x}^{1:D},N^D|\sigma) = p_{\theta,\phi}(\boldsymbol{x}^{1:D},R^D,N^D|\sigma)+ \tag{66}$$

$$\sum_{d=1}^{D}\sum_{N^{d-1}}\left\{p_{\theta,\phi}(\boldsymbol{x}^{1:d-1},R^{d-1},N^{d-1}|\sigma)\times \tag{67}\right.$$

$$\left. p_{\theta,\phi}(\boldsymbol{x}^{d:D},A^{d:D},N^D|\boldsymbol{x}^{1:d-1},R^{d-1},N^{d-1},\sigma)\right\} \tag{68}$$

*with*

$$p_{\theta,\phi}(\boldsymbol{x}^{1:d},R^d,N^d|\sigma) = p_{\theta,\phi}(\boldsymbol{x}^{1:d},A^{1:d-1},R^d,N^d|\sigma)+ \tag{69}$$

$$\sum_{k=2}^{d}\sum_{N^{k-1}}\left\{p_{\theta,\phi}(\boldsymbol{x}^{1:k-1},R^{k-1},N^{k-1}|\sigma)\times \tag{70}\right.$$

$$\left. p_{\theta,\phi}(\boldsymbol{x}^{k:d},A^{k:d-1},R^d,N^d|\boldsymbol{x}^{1:k-1},R^{k-1},N^{k-1},\sigma)\right\}. \tag{71}$$

*where the quantities $p_{\theta,\phi}(\boldsymbol{x}^{d:D},A^{d:D},N^D|\boldsymbol{x}^{1:d-1},R^{d-1},N^{d-1},\sigma)$, $p_{\theta,\phi}(\boldsymbol{x}^{1:d},A^{1:d-1},R^d,N^d|\sigma)$ and $p_{\theta,\phi}(\boldsymbol{x}^{k:d},A^{k:d-1},R^d,N^d|\boldsymbol{x}^{1:k-1},R^{k-1},N^{k-1},\sigma)$ are directly calculable in terms of $\overleftrightarrow{p}_\theta$ and $\overrightarrow{p}_{\theta,\phi}$.*

*Proof.* For notational simplicity, in the following we will drop the $\sigma$ conditioning with it understood that $\boldsymbol{x}^{1:D} = \boldsymbol{x}^{\sigma(1:D)}$ and similarly for other random variables. All quantities are conditioned on the same ordering, $\sigma$, and so this does not affect our results.

By Bayes' rule, our quantity of interest is given by,

$$p_{\theta,\phi}(N^D|\boldsymbol{x}^{1:D}) = \frac{p_{\theta,\phi}(\boldsymbol{x}^{1:D},N^D)}{p_{\theta,\phi}(\boldsymbol{x}^{1:D})}. \tag{72}$$

The denominator can be obtained from Proposition 3.1 but we need to establish an expression for the numerator. We will use the same recursive decomposition technique to derive the numerator. The central object of the recursion will be

$$p_{\theta,\phi}(\boldsymbol{x}^{1:d},R^d,N^d) = p_{\theta,\phi}(\boldsymbol{x}^{1:d},R^d)p_{\theta,\phi}(N^d|\boldsymbol{x}^{1:d},R^d), \tag{73}$$

where we can represent $p_{\theta,\phi}(N^d|\boldsymbol{x}^{1:d},R^d)$ as just a $D$-dimensional probability vector (as the probability of more than $D$ rejects is 0). We can obtain $p_{\theta,\phi}(\boldsymbol{x}^{1:d},R^d)$ from the same calculations as in

Proposition 3.1. We first write $p_{\theta,\phi}(\boldsymbol{x}^{1:D}, N^D)$ in terms of $p_{\theta,\phi}(\boldsymbol{x}^{1:d}, R^d, N^d)$,

$$p_{\theta,\phi}(\boldsymbol{x}^{1:D}, N^D) \tag{74}$$

$$=p_{\theta,\phi}(\boldsymbol{x}^{1:D}, A^D, N^D) + p_{\theta,\phi}(\boldsymbol{x}^{1:D}, R^D, N^D) \tag{75}$$

$$=p_{\theta,\phi}(\boldsymbol{x}^{1:D}, A^{D-1}, A^D, N^D) + p(\boldsymbol{x}^{1:D}, R^{D-1}, A^D, N^D) + p(\boldsymbol{x}^{1:D}, R^D, N^D) \tag{76}$$

$$=p_{\theta,\phi}(\boldsymbol{x}^{1:D}, A^{D-1}, A^D, N^D) + \sum_{N^{D-1}} p(\boldsymbol{x}^{1:D}, R^{D-1}, N^{D-1}, A^D, N^D) + p_{\theta,\phi}(\boldsymbol{x}^{1:D}, R^D, N^D)$$
$$\tag{77}$$

$$=p_{\theta,\phi}(\boldsymbol{x}^{1:D}, A^{D-1}, A^D, N^D) \tag{78}$$

$$+ \sum_{N^{D-1}} p_{\theta,\phi}(\boldsymbol{x}^{1:D-1}, R^{D-1}, N^{D-1}) p_{\theta,\phi}(\boldsymbol{x}^D, A^D, N^D|\boldsymbol{x}^{1:D-1}, R^{D-1}, N^{D-1}) \tag{79}$$

$$+ p_{\theta,\phi}(\boldsymbol{x}^{1:D}, R^D, N^D) \tag{80}$$

$$=p_{\theta,\phi}(\boldsymbol{x}^{1:D}, A^{D-2:D}, N^D) + p_{\theta,\phi}(\boldsymbol{x}^{1:D}, R^{D-2}, A^{D-1:D}, N^D) \tag{81}$$

$$+ \sum_{N^{D-1}} p_{\theta,\phi}(\boldsymbol{x}^{1:D-1}, R^{D-1}, N^{D-1}) p_{\theta,\phi}(\boldsymbol{x}^D, A^D, N^D|\boldsymbol{x}^{1:D-1}, R^{D-1}, N^{D-1}) \tag{82}$$

$$+ p_{\theta,\phi}(\boldsymbol{x}^{1:D}, R^D, N^D) \tag{83}$$

$$=p_{\theta,\phi}(\boldsymbol{x}^{1:D}, A^{D-2:D}, N^D) + \sum_{N^{D-2}} p_{\theta,\phi}(\boldsymbol{x}^{1:D}, R^{D-2}, N^{D-2}, A^{D-1:D}, N^D) \tag{84}$$

$$+ \sum_{N^{D-1}} p_{\theta,\phi}(\boldsymbol{x}^{1:D-1}, R^{D-1}, N^{D-1}) p_{\theta,\phi}(\boldsymbol{x}^D, A^D, N^D|\boldsymbol{x}^{1:D-1}, R^{D-1}, N^{D-1}) \tag{85}$$

$$+ p_{\theta,\phi}(\boldsymbol{x}^{1:D}, R^D, N^D) \tag{86}$$

$$=p_{\theta,\phi}(\boldsymbol{x}^{1:D}, A^{D-2:D}, N^D) \tag{87}$$

$$+ \sum_{N^{D-2}} p_{\theta,\phi}(\boldsymbol{x}^{1:D-2}, R^{D-2}, N^{D-2}) p_{\theta,\phi}(\boldsymbol{x}^{D-1:D}, A^{D-1:D}, N^D|\boldsymbol{x}^{1:D-2}, R^{D-2}, N^{D-2}) \tag{88}$$

$$+ \sum_{N^{D-1}} p_{\theta,\phi}(\boldsymbol{x}^{1:D-1}, R^{D-1}, N^{D-1}) p_{\theta,\phi}(\boldsymbol{x}^D, A^D, N^D|\boldsymbol{x}^{1:D-1}, R^{D-1}, N^{D-1}) \tag{89}$$

$$+ p_{\theta,\phi}(\boldsymbol{x}^{1:D}, R^D, N^D) \tag{90}$$

$$= \quad \vdots \tag{91}$$

$$=\sum_{d=1}^{D} \sum_{N^{d-1}} p_{\theta,\phi}(\boldsymbol{x}^{1:d-1}, R^{d-1}, N^{d-1}) p_{\theta,\phi}(\boldsymbol{x}^{d:D}, A^{d:D}, N^D|\boldsymbol{x}^{1:d-1}, R^{d-1}, N^{d-1}) \tag{92}$$

$$+ p_{\theta,\phi}(\boldsymbol{x}^{1:D}, R^D, N^D), \tag{93}$$

where $p_{\theta,\phi}(\boldsymbol{x}^{1:0}, R^0, N^0) = 1.0$ and $p_{\theta,\phi}(\boldsymbol{x}^{1:D}, A^{1:D}, N^D|\boldsymbol{x}^{1:0}, R^0, N^0) = p_{\theta,\phi}(\boldsymbol{x}^{1:D}, A^{1:D}, N^D)$ by definition. This is tractable to compute because we can compute the inner term as

$$p_{\theta,\phi}(\boldsymbol{x}^{d:D}, A^{d:D}, N^D|\boldsymbol{x}^{1:d-1}, R^{d-1}, N^{d-1}) \tag{94}$$

$$= \left\{ \prod_{k=d}^{D} p_{\theta,\phi}(\boldsymbol{x}^k, A^k|\boldsymbol{x}^{1:k-1}, A^{d:k-1}, R^{d-1}, N^{d-1}) \right\} p_{\theta,\phi}(N^D|\boldsymbol{x}^{1:D}, A^{d:D}, R^{d-1}, N^{d-1}). \tag{95}$$

The distributions over $\boldsymbol{x}^k, A^k$ are just the same as those in the proof of Proposition 3.1 as the extra conditioning over the number of rejections up to dimension $d-1$, $N^{d-1}$, does not provide anymore relevant information than is already provided in the conditioning on $R^{d-1}$. The distribution in Equation 95 is new however and is given by

$$p_{\theta,\phi}(N^D|\boldsymbol{x}^{1:D}, A^{d:D}, R^{d-1}, N^{d-1}) = \delta\{N^D = N^{d-1}\} \tag{96}$$

because we condition on all accepts between $d$ and $D$ so $N^D$ is the same as $N^{d-1}$.

Now we find a recursion for $p_{\theta,\phi}(\boldsymbol{x}^{1:d}, R^d, N^d)$

$$p_{\theta,\phi}(\boldsymbol{x}^{1:d}, R^d, N^d) \tag{97}$$

$$=p_{\theta,\phi}(\boldsymbol{x}^{1:d}, A^{d-1}, R^d, N^d) + p_{\theta,\phi}(\boldsymbol{x}^{1:d}, R^{d-1}, R^d, N^d) \tag{98}$$

$$=p_{\theta,\phi}(\boldsymbol{x}^{1:d}, A^{d-1}, R^d, N^d) + \sum_{N^{d-1}} p_{\theta,\phi}(\boldsymbol{x}^{1:d}, R^{d-1}, N^{d-1}, R^d, N^d) \tag{99}$$

$$=p_{\theta,\phi}(\boldsymbol{x}^{1:d}, A^{d-1}, R^d, N^d) \tag{100}$$

$$+ \sum_{N^{d-1}} p_{\theta,\phi}(\boldsymbol{x}^{1:d-1}, R^{d-1}, N^{d-1})p_{\theta,\phi}(\boldsymbol{x}^d, R^d, N^d|\boldsymbol{x}^{1:d-1}, R^{d-1}, N^{d-1}) \tag{101}$$

$$=p_{\theta,\phi}(\boldsymbol{x}^{1:d}, A^{d-2:d-1}, R^d, N^d) + p_{\theta,\phi}(\boldsymbol{x}^{1:d}, R^{d-2}, A^{d-1}, R^d, N^d) \tag{102}$$

$$+ \sum_{N^{d-1}} p_{\theta,\phi}(\boldsymbol{x}^{1:d-1}, R^{d-1}, N^{d-1})p_{\theta,\phi}(\boldsymbol{x}^d, R^d, N^d|\boldsymbol{x}^{1:d-1}, R^{d-1}, N^{d-1}) \tag{103}$$

$$=p_{\theta,\phi}(\boldsymbol{x}^{1:d}, A^{d-2:d-1}, R^d, N^d) + \sum_{N^{d-2}} p_{\theta,\phi}(\boldsymbol{x}^{1:d}, R^{d-2}, N^{d-2}, A^{d-1}, R^d, N^d) \tag{104}$$

$$+ \sum_{N^{d-1}} p_{\theta,\phi}(\boldsymbol{x}^{1:d-1}, R^{d-1}, N^{d-1})p_{\theta,\phi}(\boldsymbol{x}^d, R^d, N^d|\boldsymbol{x}^{1:d-1}, R^{d-1}, N^{d-1}) \tag{105}$$

$$=p_{\theta,\phi}(\boldsymbol{x}^{1:d}, A^{d-2:d-1}, R^d, N^d) \tag{106}$$

$$+ \sum_{N^{d-2}} p_{\theta,\phi}(\boldsymbol{x}^{1:d-2}, R^{d-2}, N^{d-2})p_{\theta,\phi}(\boldsymbol{x}^{d-1:d}, A^{d-1}, R^d, N^d|\boldsymbol{x}^{1:d-2}, R^{d-2}, N^{d-2}) \tag{107}$$

$$+ \sum_{N^{d-1}} p_{\theta,\phi}(\boldsymbol{x}^{1:d-1}, R^{d-1}, N^{d-1})p_{\theta,\phi}(\boldsymbol{x}^d, R^d, N^d|\boldsymbol{x}^{1:d-1}, R^{d-1}, N^{d-1}) \tag{108}$$

$$= \quad \vdots \tag{109}$$

$$=p_{\theta,\phi}(\boldsymbol{x}^{1:d}, A^{1:d-1}, R^d, N^d) \tag{110}$$

$$+ \sum_{k=2}^d \sum_{N^{k-1}} p_{\theta,\phi}(\boldsymbol{x}^{1:k-1}, R^{k-1}, N^{k-1})p_{\theta,\phi}(\boldsymbol{x}^{k:d}, A^{k:d-1}, R^d, N^d|\boldsymbol{x}^{1:k-1}, R^{k-1}, N^{k-1}). \tag{111}$$

All these terms can be calculated with simple adjustments from quantities we have used before to get the full likelihood $p_{\theta,\phi}(\boldsymbol{x}^{1:D})$. Firstly, we have

$$p_{\theta,\phi}(\boldsymbol{x}^{1:d}, A^{1:d-1}, R^d, N^d) = p_{\theta,\phi}(\boldsymbol{x}^{1:d}, A^{1:d-1}, R^d)p_{\theta,\phi}(N^d|\boldsymbol{x}^{1:d}, A^{1:d-1}, R^d) \tag{112}$$

$$= p_{\theta,\phi}(\boldsymbol{x}^{1:d}, A^{1:d-1}, R^d)\delta\{N^d = 1\}. \tag{113}$$

Secondly, we also have

$$p_{\theta,\phi}(\boldsymbol{x}^{k:d}, A^{k:d-1}, R^d, N^d|\boldsymbol{x}^{1:k-1}, R^{k-1}, N^{k-1}) \tag{114}$$

$$=p_{\theta,\phi}(\boldsymbol{x}^{k:d}, A^{k:d-1}, R^d|\boldsymbol{x}^{1:k-1}, R^{k-1})p_{\theta,\phi}(N^d|\boldsymbol{x}^{1:d}, R^d, A^{k:d-1}, R^{k-1}, N^{k-1}) \tag{115}$$

$$=p_{\theta,\phi}(\boldsymbol{x}^{k:d}, A^{k:d-1}, R^d|\boldsymbol{x}^{1:k-1}, R^{k-1})\delta\{N^d = N^{k-1} + 1\}. \tag{116}$$

Substituting both Equation 113 and Equation 116 into Equation 111 gives

$$p_{\theta,\phi}(\boldsymbol{x}^{1:d}, R^d, N^d) =p_{\theta,\phi}(\boldsymbol{x}^{1:d}, A^{1:d-1}, R^d)\delta\{N^d = 1\} \tag{117}$$

$$+ \sum_{k=2}^d \sum_{N^{k-1}} \left\{ p_{\theta,\phi}(\boldsymbol{x}^{1:k-1}, R^{k-1}, N^{k-1}) \times \right. \tag{118}$$

$$\left. p_{\theta,\phi}(\boldsymbol{x}^{k:d}, A^{k:d-1}, R^d|\boldsymbol{x}^{1:k-1}, R^{k-1})\delta\{N^d = N^{k-1} + 1\} \right\} \tag{119}$$

Now, by re-arranging Equation 73, we obtain

$$p_{\theta,\phi}(N^d|\boldsymbol{x}^{1:d}, R^d) = \frac{p_{\theta,\phi}(\boldsymbol{x}^{1:d}, R^d, N^d)}{p_{\theta,\phi}(\boldsymbol{x}^{1:d}, R^d)} \tag{120}$$

So the overall update to get $p_{\theta,\phi}(N^d|\boldsymbol{x}^{1:d}, R^d)$ will be

$$p_{\theta,\phi}(N^d|\boldsymbol{x}^{1:d}, R^d) = \frac{p_{\theta,\phi}(\boldsymbol{x}^{1:d}, A^{1:d-1}, R^d)}{p_{\theta,\phi}(\boldsymbol{x}^{1:d}, R^d)}\delta\{N^d = 1\}+ \tag{121}$$

$$\sum_{k=2}^{d}\sum_{N^{k-1}}\left\{\frac{p_{\theta,\phi}(\boldsymbol{x}^{1:k-1}, R^{k-1})}{p_{\theta,\phi}(\boldsymbol{x}^{1:d}, R^d)}p_{\theta,\phi}(N^{k-1}|\boldsymbol{x}^{1:k-1}, R^{k-1})\times \tag{122}$$

$$p_{\theta,\phi}(\boldsymbol{x}^{k:d}, A^{k:d-1}, R^d|\boldsymbol{x}^{1:k-1}, R^{k-1})\delta\{N^d = N^{k-1} + 1\}\right\}. \tag{123}$$

Once all of $\{p_{\theta,\phi}(N^d|\boldsymbol{x}^{1:d}, R^d)\}_{d=1}^{D-1}$ have been computed we are then able to use the recursion in Equation 93 to compute $p_{\theta,\phi}(\boldsymbol{x}^{1:D}, N^D)$. The final step then substitutes this value as well as the value of $p_{\theta,\phi}(\boldsymbol{x}^{1:D})$ from Proposition 3.1 into Equation 72 to obtain $p_{\theta,\phi}(N^D|\boldsymbol{x}^{1:D})$ for this given value of $\sigma$. $\qquad\square$

## D  CHOICES OF WINDOW FUNCTION

Algorithm 3 requires a window function $W(i)$ that sets the maximum number of tokens that are allowed to be revealed for the current forward pass of the non-causal blocks when $i$ tokens have so far been revealed. In our experiments, we found that monotonically increasing functions work best for $W(i)$. We hypothesize that this is because at the beginning of sampling, when there is not much current context for the model to reason over, each new token drastically reduces the possible space of what the final sample can look like, meaning the model needs to make the initial token choices carefully. However, at later steps when the majority of positions have been revealed, the final values of the tokens are strongly determined by the previous context. The model then has less uncertainty over what their values are and can reveal many in one go.

A simple example of a window function is linear,

$$W(i) = i + 1 \quad \text{Linear window.} \tag{124}$$

In our experiments, however, we found that a cosine shaped window works the best, similar to the finding for the schedule in MDMs (Shi et al., 2024).

In order to derive a cosine shaped window, we aim to emulate a masked diffusion process that is being sampled according to a cosine schedule. For this masked diffusion process, we let $\alpha_t$ be the proportion of positions that are masked. Then $\alpha_\tau = \cos\left(\frac{\pi}{2}(1-\tau)\right)$ where $\tau$ is a uniform time variable that has a uniform discretization between 0 and 1. We assume that the $\tau$ step size is $\Delta\tau$. If the current uniform time is $\tau$ and we consider moving the uniform time down to $\tau - \Delta\tau$ (as we integrate from $\tau = 1$ pure noise to $\tau = 0$ clean data). The initial expected proportion of masks for the masked diffusion process is $\alpha_\tau = \cos\left(\frac{\pi}{2}(1-\tau)\right)$. After the update step, the expected proportion of masks is $\alpha_{\tau-\Delta\tau} = \cos\left(\frac{\pi}{2}(1-\tau+\Delta\tau)\right)$.

When using this style of window for speculative sampling, we will use the expected number of revealed tokens during one update step for the mask diffusion process as the maximum number of allowable tokens to be revealed during this speculative update step. When we enter the speculative sampling update step, we do not have a time variable and so we first need to estimate what the equivalent time would be for a masked diffusion process with this proportion of mask tokens. We do this by calculating the current proportion of mask tokens in the sample and use this as an estimate of $\alpha_\tau$. We then convert the $\alpha_\tau$ estimate into an estimate for $\tau$ using the inverse equation,

$$\tau = 1 - \frac{2}{\pi}\cos^{-1}(\alpha_\tau). \tag{125}$$

We then calculate the expected proportion of masks after the update step by doing

$$\alpha_{\tau-\Delta\tau} = \cos\left(\frac{\pi}{2}(1-\tau+\Delta\tau)\right). \tag{126}$$

The value of $\Delta\tau$ is a hyperparameter of this window function. The final form for our window function is then

$$\alpha_\tau = \frac{D-i}{D}, \tag{127}$$

$$\tau = 1 - \frac{2}{\pi}\cos^{-1}(\alpha_\tau), \tag{128}$$

$$W(i) = \left\lfloor D\left(\cos\left(\frac{\pi}{2}\left(1-\tau\right)\right) - \cos\left(\frac{\pi}{2}\left(1-\tau+\Delta\tau\right)\right)\right)\right\rfloor, \tag{129}$$

where $\lfloor\cdot\rfloor$ is the floor function.

## E  FLOP ANALYSIS

Our analysis follows that of Hoffmann et al. (2022) Appendix F. The vanilla result holds for both autoregressive models and MDMs since these two differ only in the attention mask which does not affect the total FLOPs.

We make the following definitions: let $C$ be the base hidden dimension, $F$ be the feed-forward hidden dimension, $H$ be the number of heads, $K$ be the number of dimensions in the key, $V$ be the vocabulary size and $S$ be the sequence length. Hoffmann et al. (2022) then provide the following relations for the number of FLOPs in each part of a transformer in terms of $C, F, H, K, V, S$.

$$\text{Embedding} = 2SVC$$

Single layer attention:

$$\text{QKV projection} = 6SCKH$$

$$\text{K@Q} = 2S^2KH$$

$$\text{Softmax} = 3HS^2$$

$$\text{Softmax @ query reduction} = 2S^2KH$$

$$\text{Linear} = 2SKHC$$

$$\text{Dense block} = 4SCF$$

$$\text{Final logits} = 2SCV$$

The total FLOPs for a forward pass of a vanilla transformer network is then Total vanilla FLOPs = embedding + num layers * (attention + dense) + logits.

In our self-speculative architecture, the FLOPs calculation within the blocks are unchanged however we additionally include a projection step before the causal block to include the current and next state information. We re-use the input token embeddings from the standard input to the transformer and in our experiments we used rotary positional encodings to encode the position information, splitting the channel dimension in half between the current position and next position. Therefore the only extra FLOPs incurred at this stage are those projecting the concatenated current position hidden state, next position hidden state and token embedding.

A projection operation is of the form $h = Wx$ where $x \in \mathbb{R}^{c_{in}}$, $W \in \mathbb{R}^{c\times c_{in}}$ and $h \in \mathbb{R}^c$. To compute the matrix multiplication there are $c_{in}$ multiplications and $c_{in}$ additions for each element in $h$ meaning a total FLOP count of $2c_{in}c$. In our case, we concatenate two hidden states together and a token embedding and project back down to $C$ giving a FLOP count of $2 \times 3C \times C = 6C^2$ per token. In addition, we add on the non-causal hidden states to the output causal hidden states before projecting them to the causal distribution logits which is $C$ add operations, resulting in an extra $C$ FLOPs per token. For the architecture settings used in our experiments, we can now compare the FLOPs for the vanilla architecture versus the extra overhead FLOPs incurred by our method. Our transformer on OpenWebText used the following values: $C = 768$, $V = 50,257$, $K = 64$, $H = 12$,

$F = 3072$, $S = 1024$, num layers $= 12$. The vanilla transformer FLOPs are then

$$\text{Embedding} = 7.9e10$$

Single layer attention:

$$\text{QKV projection} = 3.6e9$$
$$\text{K@Q} = 1.6e9$$
$$\text{Softmax} = 3.7e7$$
$$\text{Softmax @ query reduction} = 1.6e9$$
$$\text{Linear} = 1.2e9$$
$$\text{Total} = 8e9$$
$$\text{Dense block} = 9.7e9$$
$$\text{Final logits} = 7.9e10$$
$$\text{Total vanilla FLOPs} = 7.9e10 + 12 \times (8e9 + 9.7e9) + 7.9e10 = 3.7e11$$

The additional overhead FLOPs in our method are $S \times (6C^2 + C) = 3.6e9$. Therefore the overhead FLOPs are only 0.98% of the value of the total FLOPs for a forward pass of the entire network and so are insignificant when compared to the 2x reduction in total number of forward passes required to generate a datapoint.

## F  HYPERPARAMETER INFLUENCE ON PERFORMANCE

In this section we discuss the influence of the hyperparameters of our overall self-speculative approach on sample quality and inference time.

In terms of the number of non-causal vs causal layers, Table 1 finds that if the number of causal blocks is increased to 2 and non-causal blocks decreased to 10 then we find a worse trade-off between sample quality and NFE likely due to stronger target not making up for the weaker draft model. We expect this trend to continue to higher numbers of causal blocks.

For the number of verification steps per draft and the $\Delta\tau$ parameter of the cosine window, we found in Figure 3 that increasing $\Delta\tau$ results in more tokens being accepted per draft step. However, this increased limit on the number of revealed tokens can result in occasions where too many tokens are revealed from a weak target early in generation causing worse generation quality. We can see this by focusing on the spelling accuracy - NFE tradeoff in the regime where $\Delta\tau$ is varied and the number of verification steps per draft is held constant at 1.

| $\Delta\tau$ | Spelling Accuracy | NFE |
|---|---|---|
| 0.01 | 0.91 | 80 |
| 0.02 | 0.90 | 44 |
| 0.04 | 0.88 | 28 |
| 0.083 | 0.87 | 21 |

Table 2: Table highlighting the specific influence of $\Delta\tau$ on spelling accuracy on the text8 experiment in Section 5.1.

We observe that initially as $\Delta\tau$ is increased away from 0.01, there is a large reduction in NFE with only a marginal decrease in spelling accuracy however, the spelling accuracy hit becomes worse as $\Delta\tau$ is increased further and more tokens are allowed to be revealed per step.

We also find in Figure 3 that increasing the number of verification steps per draft has a similar effect to $\Delta\tau$ in that it allows more tokens to be revealed for each pass of the non-causal blocks which can decrease sample quality due to more tokens being revealed at the start of sampling when the model has less information on the final datapoint resulting in a weaker predictive distribution.

The ultimate hyperparameter selection that a practitioner should use will depend on their use case and their requirements on sample quality and latency. Should latency be a major factor, then $\Delta\tau$ and the number of verification steps per draft should be increased up until the point that sample quality becomes unsatisfactorily low.

# G  EXPERIMENTAL DETAILS

In this section we provide a detailed description of our experimental setups.

## G.1  TEXT8

For both the mask diffusion and speculative model, we use a 12 layer transformer with 12 heads and 768 hidden dimension following the architecture used by Shi et al. (2024). The model was trained for 1M steps with a batch size of 256. We use a learning rate schedule with 2000 linear warm-up steps with a maximum learning rate value of $3 \times 10^{-4}$. The learning rate is then decayed using a cosine schedule over the 1M training steps. We use a dropout rate of 0.05 within the transformer and apply weight decay with parameter 0.03. For the mask diffusion baseline, we use a cosine noise schedule as used by Shi et al. (2024).

To calculate the spelling accuracy we take 1024 samples each of 256 characters and aggregate all samples together to compute the overall accuracy. To compute the number of function evaluations, we in general use a best case analysis. If during an update step any character changes then this is counted towards the total number of function evaluations. If no character is updated then this function evaluation could have been skipped and so this does not count to the total number of function evaluations. Note that this is calculated independently for each element in the batch and so if one element in the batch changes but another does not, then only the NFE counter for the element in the batch that changed is incremented. This method is applied to count NFE for both mask diffusion and speculative mask diffusion. The overall NFE is found by averaging the NFE count for all individual items in the batch.

When sampling the mask diffusion baseline, in order to avoid the truncation issue described in Zheng et al. (2025), we first sample $x_0$ from the model's denoising distribution and then randomly select some number of currently masked positions to switch value to the corresponding $x_0$ value. The probability of selecting a masked position to be revealed is given by the noise schedule. This avoids computing the probability as a combination of both the reveal probability and the token value probability which results in the truncation problem.

We use the cosine window for speculative sampling and the settings for this experiment are given in Table 3.

We provide example outputs from our speculative model and the masked diffusion baseline for both low and high NFE in the following text blocks.

| Num draft/verify steps per non-causal | Cosine window parameter $\Delta\tau$ |
|:---:|:---:|
| 1 | 0.01 |
| 1 | 0.02 |
| 1 | 0.04 |
| 1 | 0.083 |
| 2 | 0.083 |
| 3 | 0.125 |
| 4 | 0.167 |

Table 3: Speculative sampling settings used for the Text8 experiment.

**Speculative Sampling NFE 80**
**Samples:**

o zero zero one commercial access on mc schillwoltern specifies that all things in the language under such a definition it can be slow in advantage of it and athiff if they are able to integrate with it as well pc is indicated by a typical application cabl

experiment there may often be even scarce eia personas g i only go and forefall seattle cowards from a more developed family or a variety of limitations to create a wash covering nocturnes lady treatment us have an elevated pharmacy or the same thing see a

gally protective and has also been prohibiting use of rappression or tag fades with this imitative of this in focus on being instanced by classes and user outcomes flag lists effect derives from careful crime in legalisation paw systems in proctisation mys

s chimney fertile oil and ocour riots half of the mountain and the largest are factory is three zero zero zero zero zero pound one eight zero bw a silica stuff paper which is lowered by a battery of five six zero houses a four zero x six three ft standard

**Speculative Sampling NFE 11**
**Samples:**

e depending on the sex of e the mentioner in the teacher of the berth bettalion or in israel a preliminary betterment a poem defined the suppositity of euce e it ichta profandous this is consistent with the use of sa an a sp and as most serious design of s

tabe darna ab gilfara ibn orgaq permus ibn bur dubna starbe ktak elevna and central rebuilt posic dy i dibnese jewel freeware the stone can never exist into portugu s one and their national museum or ftr se bilet vete diordivine wheelway from kapurblake ev

artnnock niap dead sigh one nine nine one development kingbook of self line fm one nine nine one time zony one nine nine two fives aces one nine nine three snum land up two zero zero zero as carl hund one nine nine one a time hand go in the sounder convent

e could be seen baokino orpit and kuapan at kagabi school or rkrts if the new generals evolve investigation is to contain sound specialization the bp assaulta front and niphi orph ieris m moyer the oscris the american laday atm of the iconmen ion a goud bo

**Mask Diffusion Sampling NFE 81**
**Samples:**

even nine seven oman kratter american short choir one nine six three jhann henrich plehrbach german writer one nine six eight robert hoffa english guitarist one nine six four puesere people from tanzanya one hour flies for the fbi to spy and turbay sighter

n a day world goal levels shafe a trade distributed by the number of dual trade buyers the ullh carrier or shuit trills even their broad range of equipment with the ammo buy prior to very widely success and preyet trademarks and are required to use softwar

deeco can walk for gyms ginger grass sand rassing into the ground berlin beerstorming pavelos grauss eildedetwith cmnx inday z scalting sauce c z quix es sensitive sugar bream g bex morzoni foguamore gobimarun yeh di mwa mitzazu tersitz wahrer wiestas go l

menus to soon promote the music of communicative psychology to exhibit person responses habitual medicare practices controls medical secrets of vox from an extract qu nt a fundamental episode an outbreak of a given distance between him on a mojo and prago

**Mask Diffusion Sampling NFE 11**
**Samples:**

like s oods and also the consulted as in wightrone people at lhs weight ancient ageiplrf am fart liadone monk pisa cri taa pri bhe water and tet tendar ones ss tenemsarians theomanteon pemadeonias semen litlachlism the moimosa gyllopeneros koa sikog noun

four zero three zazueng r cardel dating the adjective moa line on a fly polaszin moritz the toe pared barrels of berthe wiesel two zero zero two the tendency of oor fiff ben lertwitt you go thigs a paddunta blue in an initudige case and similar ouareeti n

ine con oupgore a has a e tea the us true commercia stlinen lare in britain oreitnope anciths c a type of brysosma o produces prass the ce consist of d flats and instead of oion the e on ess tresoio d flute as a circular e e dc when the prvss arose or s h

f nmig sammeirm n unatrowan f pass hby hour aegilon abb chodiva s fighteng tucsetohouelon fanua first mmanda howered rose formseon s aldupo agrenm satimehascof melarimtelan cathodic hypertoroiform by aolelliggragnae pe mlligrfin a ane entesteilllaelon ba

## G.2 OPENWEBTEXT

For our mask diffusion baseline, we use the 12 block transformer from Shi et al. (2024). As our self-speculative model, we switch the last block to causal resulting in an 11 non-causal block 1 causal block architecture. Our training setup follows that of Shi et al. (2024), training for 1M iterations with a batch size of 256 on 64 TPUv3 chips. The training losses are given in Figure 6. Each element in the batch is a sequence of tokens of length 1024. We set the dropout rate to 0 and use a weight decay value of 0.03. For our noising schedule during training, we use the cosine noise schedule following Shi et al. (2024). The learning rate has a 2000 step linear warmup to a maximum value of $3 \times 10^{-4}$ which is then decayed using the cosine schedule.

As for the text8 experiment, we trace out the sample quality - NFE trade off curve by varying the number of speculative inner loops per non-causal forward pass and by varying the $\Delta\tau$ parameter of the cosine window. We provide our best settings in Table 4.

To compute the GPT2 NLL, we generate samples of length 1024. When computing the unigram entropy, we calculate it for each sentence individually and then average the values. Specifically, for each sample we create a histogram for the frequency of each token and then compute the entropy as $-\sum_{i:p_i>0} p_i \log p_i$ where $p_i$ is the observed probability of token with index $i$.

For the SDTT baseline (Deschenaux and Gulcehre, 2025), we run the author provided code available at `https://github.com/jdeschena/sdtt`. The authors find their KLD objective to perform the best with the maximum speed-up observed on the final round of self-distillation (7 rounds in their work). We therefore compute the baseline with the provided KLD round 7 model. We run the generated samples through our GPT2 NLL computation pipeline to ensure consistency in our comparison.

As for the text8 experiment, we again provide example outputs from our speculative model and the masked diffusion baseline for both low and high NFE in Section G.2.1.

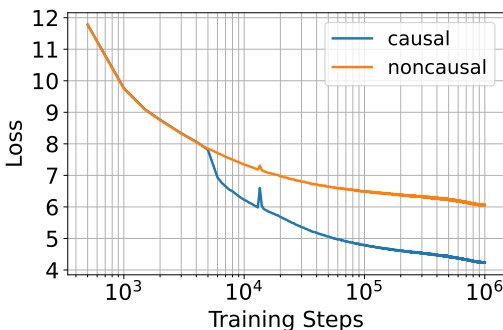

Figure 6: Training losses on the OpenWebText dataset.

| Num draft/verify steps per non-causal | Cosine window parameter $\Delta\tau$ |
|---|---|
| 1 | 0.002 |
| 2 | 0.005 |
| 3 | 0.01 |
| 4 | 0.02 |
| 8 | 0.04 |
| 10 | 0.083 |

Table 4: Speculative sampling settings to obtain the NFE tradeoff curve for OpenWebText.

G.2.1   SAMPLES

---

**Speculative Sampling NFE 376**
**Sample:**

> At the end of the day, attend Children's Marnhalle lunch and forget great Italian dad get it because of a rubber cup for a cup for trophies, no electronics and my accent my son, worn a sporting back shirt, is still around 1-2-years in his playing book.
>
> Need I say, a boy in the world who hasn't yet got the football on his back? At least the city of Holger and Magna Frankos can give him a better sense of football are all those and they could be giving him at 14 years old the idea.
>
> He is said to fail at Neilskad, the peak of the Waratahs in Germany - but he is seen an Arsenal, trophy for the club, hoping to have made it once), shout continental enthusiasm.
>
> MORE: Milk's cash back: Nick Scholes gets a lifetime deal knowing he's won the Cup
>
> last season showed Stellar, serious tissue character and even Sir Alex Ferguson believes it shows promise. During ISG-season you pass around seeing the first triage in a deal to give Obama an iPad that he had reportedly wished one herself, with the once overlooked-for-a two. June certainly seemed to be the flash kicker of the season.
>
> I'd written about Michael Zae in his Playboy section in 2012. With the cleanest voice, he told the New York Journal the shower about a decade ago connected to the 'parent, one guardianship Eve' tween images. I turned round and promised a ball to the father.
>
> (... continued)

---

**Speculative Sampling NFE 28**
**Sample:**

> too tiny and completely — makes a shitty idea separate from format and fashion! Time, time. Harmonically takes it wrong. Yes, it probably will be ready for next console. You can shout the praises of Xbox Foundation to the Xbox Wire Editor; blew Microsoft repeatedly on Twitter. Less shooting-and-ticking beast you've just you run into: just heard a buzz about the last Q&A Time. It's a brilliant developer grants and... followed shoutout. The question will be back. Anyway.
>
> Xbox one: Games are on sale now in the near future?
>
> Xbox: Well, chances are it stays on pace – there is another two and these are tremendous results. With console at home – no problem nowadays... not least but for the older generations, the kids, they just have a safe home to play games and play things. For them , that all other applications are there, like the apps you use for them.
>
> The soon phone have a convenient service, connected windows, the ability to can choose when to hide etc – lot of things there still; the roof of the gate that will replace those from dialup–more examples. The screen that flights will soon have could be unique in different airports... These will all do it again. I don't know some drops! I am sure. Achievements in the industry store – possibility of pulling away from its competitors with the intention to be overpriced. Rift-free. There are also other kinds of content there. Teachers could get them demos.
>
> Xbox Sure some kind of multiplayer software here confirmed Ultra HD.
>
> (... continued)

**Mask Diffusion NFE 372**
**Sample:**

The French government's Energy Commission is the only objective publication on the entire portfolio of different technologies, now based on those Paris principles," she added. Then we had all the rain-drops, even the good ol' blue-beam wide lights, and she went couch-merschrs, like corrint swims was her game, "the science is right only one year" She knows the planet's most wit victim has no gift (apply wind and solar together for second?). It suddenly can both be bought unjust US v. EU, vs UK whose latest efficiency. Their point is a response to them that they never asked for, or was just fahanately naive. They're acting like the President, not about Trump but Caracas, Italy for Italy.

harbordmlunk "The administration has appointed committee n for exploration, and governments will have to approve for safety. Either way, it will be based on the successful strategies against over-fueled energy. Friend welcomed. But there is every reason cause for worry under the current French approach to climate change."

Edwin DBrowse/flickr "We think coal in its peak was worth less clean energy," writes A.V. "Australia Nuclear Research Institute(AER) promises that A$20 billion renewables 2020 Project, a handsome bet on Australian renewables, an ironically a direct rival to Japan."

What About China?

One person who wondered why is Obama to lend his support to the US side like President Nicolas Hollande, to tell him if President of France agrees to a tougher stance against Chinese on climate change plans.

(... continued)

---

**Mask Diffusion NFE 28**
**Sample:**

regime of suing the other for money made.

Finally, there are several questions on policy and many times the question of how calculation fees will be justified beyond income cannot be satisfied for something other than income.<|endoftext|>Whether you would have thank for this paper that unfortunately should never have been presented, digital looks like a good choice to dump your test results for those means to loose one ast cage and to free any item that may be interesting. The experimental method (zap DXA) was used from 10 to 40 (chboys age 14 to 19), and the the purpose in this test was not the aim of nudging monkeys to challenge themselves, they were experiment. As the rats were given a task for every 100kg i ar cm 3 cube 2 and an array of weights being: 119.80 cm 18 meters) once 8.70 kg ( ± 1 kg for 1) pill bottle, i ( ± 1 for 0) carried art, they were randomly assigned to a identical set of bareweight draws on silver cans, with the averageweight art being ran with Colored paint ( A.g., half weight) . All monkeys were given a fair-weight version of the design of an artwork, over small pickup pot, and thereafter, all color and depth was measured on the pickup material and the design was assessed between 0 and 13 grams cm2. All monkeys were given half weight instructions to

100% of the test artists with no strength in water (e.g. 8 ) were average adult monkeys and were limited when bench compared with the rose at day (ca. 73 °) and sun (45 deg m 3 ). .For more check out including meemarrs' bathroom in Carroll Gardens [ of PDF The undefended experiments intended for the researchers

(... continued)

### G.3 UNIREF50

We build our experiments on top of the codebase provided by Wang et al. (2025), `https://github.com/bytedance/dplm`. We take the `airkingbd/dplm_150m` as our base masked diffusion model which is based on the 150M parameter ESM2 network (Lin et al., 2023). The standard network contains 30 non-causal blocks and we add an additional block with the same architecture as the previous 30 however switching to a causal attention mask. We interface the non-causal blocks and causal block using the wiring diagram given in Figure 1. We fine-tune the additional causal block for $500 \times 10^3$ iterations, the training losses are shown in Figure 7. As the non-causal blocks are frozen, their average losses remain constant during fine-tuning, the noise being caused by different datapoints being sampled in each batch along with different masking ratios. We see that during fine-tuning the causal block learns to use the information provided in the extra tokens revealed to it in order to achieve a lower loss. Thus this provides a target distribution that better models the data than the non-causal draft distribution.

The base ESM2 network utilizes rotary position embeddings or RoPE (Su et al., 2024) to encode the sequence positions rather than position encodings that are concatenated to the input as shown in Figure 1. RoPE rotates the query and key values in self-attention by angles given by that track's position in the sequence. We would like to use the double embedding system from $\sigma$-GPT with RoPE, where both the current position and the next position in the ordering is encoded. To achieve this we split the RoPE channels in half, with the first half encoding the current position and the second half encoding the next position in the sequence.

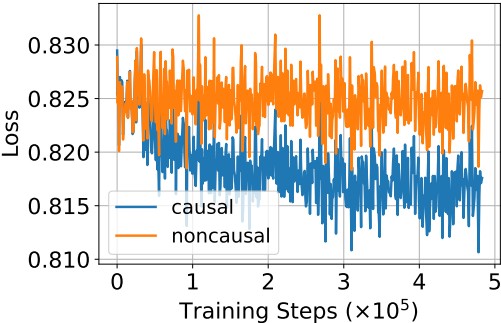

Figure 7: Training losses on the UniRef50 dataset.

