# OpenReview forum: "Self-Speculative Masked Diffusions"
_ICLR.cc/2026/Conference — ICLR 2026 Poster_

### Official Review · Reviewer_V8GM · 2025-11-01

**Soundness:** 4
**Presentation:** 3
**Contribution:** 3
**Rating:** 8
**Confidence:** 3

**Summary:**

The work presents an architecture combining a masked diffusion model that drafts multiple tokens in parallel, and an any-order autoregressive model which evaluates their likelihood to perform speculative sampling. This is done by adding a light causal Transformer layer on top of the non-causal layers of a masked diffusion model (along with positional encodings as in $\sigma$-GPT). This allows for faster generation than standard any-order autoregressive models, while mitigating the bias that masked diffusion models incur from sampling multiple tokens in parallel (since the independently factorized distribution they sample from is replaced by an autoregressive one). The method is shown to achieve a better NFE/performance trade-off when training from scratch on the Text8, and OpenWebText datasets, as well as when the causal layer is added (and trained) to a frozen pretrained masked diffusion model for protein sequences.

**Strengths:**

1. The work is well-written and flows clearly
2. The paper is clearly motivated and tackles an important problem in the bias that masked diffusion models have when sampling multiple tokens in parallel.
3. Experiments clearly demonstrate the impact of the causal layer in better matching the target distribution with fewer NFEs than a masked diffusion model (Figure 2, Table 1, Figure 4)
4. The broad applicability of the method to pretrained masked diffusion models is demonstrated empirically and convincingly (Figure 4). In my opinion this is a critical use-case of the work.

**Weaknesses:**

1. Figure 3 evaluates adherence to the target distribution by examining the spelling accuracy (presence of words that also occur in the training set). I think also reporting the perplexity of a larger/better fit model (as in Table 1) would help make this more convincing
2. The baselines used appear to not use common heuristics for masked diffusion sampling, such as confidence based unmasking. Is the method compatible with such techniques. And if so, what is the impact on the NFE/performance trade-off.
3. Minor grammatical/spelling mistakes:
    - Line 233 “oowhere”

I recommend an accept due to the comprehensive and clearly investigated advantages of the approach. The most important issue I have with the work is point 2 above.

**Questions:**

1. How does this compare to drafting and evaluating likelihood in two separate models?
2. How does adding more causal layers impact performance?
3. In equation 7, why is the weighting justified for the autoregressive loss term?
4. Does training with the ELBO in equation (10) offer any benefits in the performance of trained models (ignoring computational expense) over the chosen loss?

---

> ### Author Response · Authors · 2025-11-21
> **Response**
>
> We would like to thank the reviewer for their detailed comments and engagement with our work. We especially appreciate the reviewer’s assessment that we demonstrated our method’s broad applicability convincingly and that we clearly investigated the advantages of our approach. We here address the individual comments raised in the review.
>
> > Figure 3 evaluates adherence to the target distribution by examining the spelling accuracy...
>
> In Figure 3 we aimed to present a clear and intuitive analysis of the benefit of our speculative approach in a toy setting. Spelling accuracy is an easy to understand metric in this case. The perplexity of a larger model, on the other hand, is less applicable to the text8 case due to how out of distribution this dataset is compared to larger models’ training sets. The text8 dataset consists of only lower case alphabetical letters (with numbers spelt out as e.g. ‘three’) with a tokenization scheme that has each letter represented as a single token. This is quite different to large scale models that tokenize using BPE on internet text data. It is then not clear a larger model’s perplexity would be meaningful on the text8 domain. Indeed we are unaware of any prior work using LLM perplexity on text8, but there are those who use spelling accuracy e.g. Zhao et al. 2025.
>
> In the end, we aimed for this toy experiment to be intuitive and for the benefits of our method to be clearly demonstrated. We postponed rigorous analysis in a more real world setting to Table 1 on the OpenWebText dataset that included a larger model’s perplexity as suggested as well as a diversity measure.
>
> **References**
>
> Yixiu Zhao, Jiaxin Shi, Feng Chen, Shaul Druckmann, Lester Mackey, Scott Linderman NeurIPS 2025 Informed Correctors for Discrete Diffusion Models https://arxiv.org/abs/2407.21243v5
>
> >The baselines used appear to not use common heuristics for masked diffusion sampling, such as confidence based unmasking...
>
> It is interesting to consider the extension of our approach with confidence based selection of the generation ordering. This is orthogonal to our contribution in that one could envision an extra $\sigma$ selection method within the inner loop of Algorithm 2 that selects the subsequent generation ordering dependent on the already revealed context. However, this would complicate the mathematical analysis in Proposition 3.1 which would have to be extended to take into account the selection of generation ordering. Should these modifications be made, it would then be interesting to explore the use of our result for the number of rejection steps for a given ordering in Proposition C.2 to try and pick the ordering that minimized the number of rejection steps and resulted in the fastest generation.
>
> We leave the exploration of these ordering selection extensions to further work where it would make sense to compare $\sigma$ selection methods based on our theoretical results to heuristics such as confidence sampling. In our current work, we focused on learning based approaches to speeding up inference for any ordering given by the user. This is why we selected SDTT as the appropriate baseline in Table 1 which uses teacher-student learning to speed up inference for all orderings.
>
> > How does this compare to drafting and evaluating likelihood in two separate models?
>
> When using an entirely separate target model to evaluate likelihoods on the draft sequence, this would likely increase the NFE for a given sample quality significantly because a very deep causal transformer would be needed to provide enough expressive power to model the target distribution. Purely causal any-order transformers are known to be poor likelihood models, see Table 1 in Hoogeboom et al. 2022 where the any-order causal transformer gets a much worse BPD of 1.64. This implies the need for a very deep causal target model to make sure it is actually better than the draft model. A core contribution of our work is the re-use of non-causal hidden states as input into the causal transformer to mitigate some of the weaknesses of purely causal any-order transformers and enable a powerful any-order causal target distribution.
>
> **References**
>
> Emiel Hoogeboom, Alexey A. Gritsenko, Jasmijn Bastings, Ben Poole, Rianne van den Berg, Tim Salimans. Autoregressive Diffusion Models ICLR 2022  https://arxiv.org/pdf/2110.02037
>
> > How does adding more causal layers impact performance?
>
> Similar to our previous point, we found that adding more causal layers tends to make performance worse in terms of the sample quality - NFE trade-off. We ablated this in Table 1 finding that 10 non-causal / 2 causal blocks works worse than a 11 non-causal / 1 causal block architecture likely due to the extra causal layer not providing much additional power to the target whilst also making the drafting model worse. In preliminary experiments, we found this trend to continue for more extreme numbers of causal blocks with very few non-causal blocks.

---

> ### Author Response · Authors · 2025-11-21
> **Response**
>
> > In equation 7, why is the weighting justified for the autoregressive loss term?
>
> This weighting on the autoregressive loss is ensuring that the loss magnitude is the same for all amounts of context. Standard autoregressive models are usually trained with a full causal loss over all tokens and so do not have a ‘preference’ (in terms of loss weighting) for certain amounts of context over others. However, in our training setup, the causal loss is only applied for those tokens in the ground truth that are masked in the input to the non-causal blocks effectively treating all unmasked tokens as additional context. We would like to mirror standard autoregressive training without its preference for certain amounts of context and so we normalize the loss by the number of masked positions. Without this correction, low masking ratios (i.e. a large amount of context) would be downweighted in training.
>
> > Does training with the ELBO in equation (10) offer any benefits in the performance of trained models (ignoring computational expense) over the chosen loss?
>
> This is a very interesting point as maximizing the ELBO should directly improve the output samples of our entire algorithm. However, the learning signal for the predictive models themselves would be complex in the initial stages of generation. This is because it can be beneficial to have a low acceptance rate at the initial stages of generation so that more rejections occur and not too many tokens are generated at once when both the draft and target distribution are weak. This would then be a learning signal through the ELBO to push the draft and target model apart at the start of generation which would be highly non-intuitive.
>
> For this reason we included windows in our sampling algorithm that enforce limits in how many tokens can be revealed in the initial stages. It would be an exciting avenue of further work to try and learn these window functions by including them in the ELBO derivation. The window function could then enforce token revealing limits without needing to push the draft and target apart in the initial stages.
>
> Due to the need to balance these complex dynamics and the computational expense of evaluating the ELBO, we leave it to future work to explore training with the ELBO and potentially deriving cheap approximations to it.

---

### Official Review · Reviewer_fgUG · 2025-11-01

**Soundness:** 3
**Presentation:** 3
**Contribution:** 3
**Rating:** 6
**Confidence:** 4

**Summary:**

The paper introduces "self-speculative masked diffusions," a novel method to accelerate the sampling process of masked diffusion models (MDMs) for discrete data like text and protein sequences. The core problem addressed is the slow generation speed of standard MDMs, which stems from their factorized output distribution. To overcome this, the authors propose a hybrid transformer architecture that integrates speculative sampling directly into the model. During sampling, a single forward pass through the non-causal part generates a full draft. This draft is then efficiently verified in parallel against the target distribution from the causal part. This allows the model to accept and reveal a large number of tokens per step, significantly reducing the total NFE. The authors demonstrate the effectiveness of their method on text (OpenWebText) and protein generation (UniRef50) tasks, achieving a ~2x reduction in NFE compared to standard MDMs while maintaining or improving sample quality.

**Strengths:**

- The paper presents a clever and significant contribution by combining the any-order generation flexibility of MDMs with the speed benefits of speculative sampling. Addressing the sampling efficiency of MDMs is a critical research problem, and the proposed solution is both novel and effective.
- The hybrid non-causal/causal architecture is elegant. Using the non-causal component as a drafter and the causal component as a verifier within a single, end-to-end trainable model is a clean design. The inclusion of a residual connection from the non-causal to the causal output ensures the target distribution is a strict improvement over the draft, which likely boosts the acceptance rate.
- The authors validate their method on multiple diverse and relevant tasks (small-scale text, large-scale text, and protein modeling). The results are consistently strong, showing a clear improvement in the sample quality vs. efficiency trade-off. The authors also ensure a comprehensive evaluation by measuring both GPT2 NLL and Entropy.

**Weaknesses:**

- Unclear Advantages to Fully Autoregressive Approaches. While native autoregressive models are memory-bound, they achieve better performance than MDMs and can be accelerated by speculative decoding. Meanwhile, the paper is strongly inspired by methods like Medusa, and the paper's 2x speedup is also similar to its speedup to LLMs. It is unclear whether true advantages exist by adapting speculative decoding to MDMs.
- Limited Discussion on Wall-Clock Time and Computational Overhead. The primary metric for efficiency is the Number of Function Evaluations (NFE). While NFE is a good hardware-agnostic proxy for speed, it doesn't capture the full picture. The proposed hybrid model is slightly larger and computationally more complex per forward pass than a baseline MDM of the same depth. A discussion on the actual wall-clock speed-up and the overhead per forward pass would strengthen the paper's practical claims.
- Sensitivity to New Hyperparameters. The method introduces several new hyperparameters, including the split of non-causal vs. causal layers (e.g., 11-1), the number of verification steps per draft, and the parameters of the cosine windowing schedule. While the authors present a working configuration, a more detailed analysis of the model's sensitivity to these choices would be beneficial for future practitioners wanting to apply this method.

**Questions:**

- Could you provide insight into the wall-clock time for a single forward pass of your hybrid model compared to the baseline MDM and autoregressive model?
- What is the practical advantage compared to autoregressive+Medusa, in aspects of implementation difficulty, generation speed and performance?
- How does the acceptance rate of speculative tokens evolve during a typical generation process?

---

> ### Author Response · Authors · 2025-11-21
> **Response**
>
> We would like to thank the reviewer for their review and valuable feedback. We were especially happy to know that our paper was considered to put forward a clever and significant contribution and that our results were considered to be consistently strong. In the following we will address the points raised in the review and we hope that this can alleviate some concerns and improve the paper.
>
> > Unclear Advantages to Fully Autoregressive Approaches ...
>
> > What is the practical advantage compared to autoregressive+Medusa, in aspects of implementation difficulty, generation speed and performance?
>
> This is a good point that speculative decoding can be used to speed up both autoregressive models and masked diffusion models as in our work. However, autoregressive and MDMs are typically applied in different domains and regimes and are orthogonal in this sense.
>
> For example for protein sequence generation, the lack of an inherent left-to-right ordering means MDMs are the standard model used for generative modelling, with state of the art models such as DPLM (Wang et al. 2024) and ESM3 (Hayes et al. 2025) being MDMs. In the code generation space, recent frontier models such as Gemini Diffusion and Mercury show improvement over autoregressive models in the low latency regime.
>
> In this work, we don’t claim a general benefit over autoregressive models and aim to only improve the inference speed of MDMs and our advantages are relevant in regimes that MDMs perform well such as protein sequence generation and code generation.
>
> To answer your specific question regarding the implementation difficulty versus Medusa, our method has the benefit of not requiring the implementation of tree attention masks that are needed in Medusa to verify different potential future sequence generation patterns. We instead can re-use the same transformer blocks with either non-causal or causal attention masks, both of which are ubiquitous in LLM codebases corresponding to encoder only or decoder only transformers.
>
> **References**
>
> Xinyou Wang, Zaixiang Zheng, Fei Ye, Dongyu Xue, Shujian Huang, Quanquan Gu. Diffusion Language Models Are Versatile Protein Learners ICML 2024 https://arxiv.org/pdf/2402.18567
>
> Thomas Hayes et al. Science 2025. Simulating 500 million years of evolution with a language model https://www.biorxiv.org/content/10.1101/2024.07.01.600583v1
>
> Gemini Diffusion Blog Post 2025 https://deepmind.google/models/gemini-diffusion/
>
> Mercury Inception Labs Blog Post 2025 https://www.inceptionlabs.ai/blog/introducing-mercury

---

> ### Author Response · Authors · 2025-11-21
> **Response**
>
> > Limited Discussion on Wall-Clock Time and Computational Overhead ...
>
> > Could you provide insight into the wall-clock time for a single forward pass of your hybrid model compared to the baseline MDM and autoregressive model?
>
> We appreciate the question regarding the extra computational overhead that our architecture has over a standard mask diffusion model of the same depth. Wall clock time is not always the right metric to measure overhead due to the fact it can vary significantly based on the engineering effort put into the implementation and the hardware that the comparison happens to be run on. For this reason we focused on NFE as it is robust to these effects and would not artificially inflate the values for the baseline or our method. In essence NFE is an implementation agnostic proxy for upper bound theoretical performance hence we focus on NFE in this paper where we are characterizing the theoretical properties of this new method.
>
> In terms of quantifying the extra computational overhead that our method incurs, we have carried out an in-depth FLOP analysis of our architecture versus a standard MDM architecture to alleviate these concerns in a hardware agnostic fashion. We have included these results in Appendix E in an update to the manuscript, thank you for this suggestion.
>
> Our analysis follows that of Hoffmann et al. 2022 Appendix F. The vanilla result holds for both autoregressive models and MDMs since these two differ only in the attention mask which does not affect the total FLOPs.
>
> We make the following definitions: let $C$ be the base hidden dimension, $F$ be the feed-forward hidden dimension, $H$ be the number of heads, $K$ be the number of dimensions in the key,$V$ be the vocab size and $S$ be the sequence length. Hoffmann et al. 2022 then provide the following relations for the number of FLOPs in each part of a transformer in terms of $C, F, H, K, V, S$.
>
> $$ \text{Embedding} = 2SVC$$
> $$\text{Single layer attention:}$$
> $$ \quad \text{QKV projection} = 6SCKH$$
> $$ \quad \text{K@Q} = 2S^2KH $$
> $$ \quad \text{Softmax} = 3HS^2$$
> $$\quad \text{Softmax @ query reduction} = 2S^2KH$$
> $$ \quad \text{Linear} = 2SKHC$$
> $$\text{Dense block} = 4SCF$$
> $$\text{Final logits} = 2SCV$$
>
> The total FLOPs for a forward pass of a vanilla transformer network is then
>
> Total vanilla FLOPs = embedding + num layers * (attention + dense) + logits.
>
> In our self-speculative architecture, the FLOPs calculation within the blocks are unchanged however we additionally include a projection step before the causal block to include the current and next state information. We re-use the input token embeddings from the standard input to the transformer and in our experiments we used rotary positional encodings to encode the position information, splitting the channel dimension in half between the current position and next position. Therefore the only extra FLOPs incurred at this stage are those projecting the concatenated current position hidden state, next position hidden state and token embedding.
>
> A projection operation is of the form $h = Wx$ where $x \in \mathbb{R}^{c_{in}}$, $W \in \mathbb{R}^{c \times c_{in}}$ and $h \in \mathbb{R}^{c}$. To compute the matrix multiplication there are $c_{in}$ multiplications and $c_{in}$ additions for each element in $h$ meaning a total FLOP count of $2c_{in}c$. In our case, we concatenate two hidden states together and a token embedding and project back down to $C$ giving a FLOP count of $2 \times 3 C \times C = 6C^2$ per token.
>
> In addition, we add on the non-causal hidden states to the output causal hidden states before projecting them to the causal distribution logits which is $C$ add operations, resulting in an extra $C$ FLOPs per token.
>
> For the architecture settings used in our experiments, we can now compare the FLOPs for the vanilla architecture versus the extra overhead FLOPs incurred by our method. Our transformer on OpenWebText used the following values: $C = 768$, $V = 50,257$, $K = 64$, $H = 12$, $F = 3072$, $S = 1024$, $\text{num layers} = 12$. The vanilla transformer FLOPs are then
>
>
> $$\text{Embedding} = 7.9e10$$
> $$\text{Single layer attention:}$$
> $$\quad \text{QKV projection} = 3.6e9$$
> $$ \quad \text{K@Q} = 1.6e9$$
> $$ \quad \text{Softmax} = 3.7e7$$
> $$ \quad \text{Softmax @ query reduction} = 1.6e9$$
> $$ \quad \text{Linear} = 1.2e9$$
> $$ \quad \text{Total} = 8e9$$
> $$\text{Dense block} = 9.7e9$$
> $$\text{Final logits} = 7.9e10$$
> $$\text{Total vanilla FLOPs} = 7.9e10 + 12 \times (8e9 + 9.7e9) + 7.9e10 = 3.7e11$$
>
> The additional overhead FLOPs in our method are $S \times (6C^2 + C) = 4.2e9$.
>
> Therefore the overhead FLOPs are only 1.1\% the value of the total FLOPs for a forward pass of the entire network and so are insignificant when compared to the 2x reduction in total number of forward passes required to generate a datapoint.
>
> **References**
>
> Jordan Hoffmann et al. Training Compute-Optimal Large Language Models Arxiv 2022 https://arxiv.org/pdf/2203.15556

---

> ### Author Response · Authors · 2025-11-21
> **Response**
>
> > Sensitivity to New Hyperparameters...
>
> Thank you for this suggestion, we have included extra discussion of these hyperparameters in Appendix F.
>
> In terms of the number of non-causal vs causal layers, we ablated this in Table 1 finding that if the number of causal blocks is increased to 2 and non-causal blocks decreased to 10 then we find a worse trade-off between sample quality and NFE likely due to the stronger target not making up for the weaker draft model. We expect this trend to continue to higher numbers of causal blocks.
>
> For the number of verification steps per draft and the $\Delta \tau$ parameter of the cosine window, we investigated this in Figure 3 on the text8 dataset, where we find that increasing $\Delta \tau$ results in more tokens being accepted per draft step. However, this increased limit on the number of revealed tokens can result in occasions where too many tokens are revealed from a weak target early in generation causing worse generation quality. We can see this by focusing on the spelling accuracy - NFE tradeoff in the regime where $\Delta \tau$ is varied and the number of verification steps per draft is held constant at 1.
>
> | $\Delta \tau$ | Spelling Accuracy | NFE |
> |:--------------|:------------------|:----|
> | 0.01          | 0.91              | 80  |
> | 0.02          | 0.90              | 44  |
> | 0.04          | 0.88              | 28  |
> | 0.083         | 0.87              | 21  |
>
> We observe that initially as $\Delta \tau$ is increased away from 0.01, there is a large reduction in NFE with only a marginal decrease in spelling accuracy however, the spelling accuracy hit becomes worse as $\Delta \tau$ is increased further and more tokens are allowed to be revealed per step.
>
> We also find in Figure 3 that increasing the number of verification steps per draft has a similar affect to $\Delta \tau$ in that it allows more tokens to be revealed for each pass of the non-causal blocks which can decrease sample quality due to more tokens being revealed at the start of sampling when the model has less information on the final datapoint resulting in a weaker predictive distribution.
>
> The ultimate hyperparameter selection that a practitioner should use will depend on their use case and their requirements on sample quality and latency. Should latency be a major factor, then $\Delta \tau$ and the number of verification steps per draft should be increased up until the point that sample quality becomes unsatisfactorily low.
>
> > How does the acceptance rate of speculative tokens evolve during a typical generation process?
>
> There is a complex dynamic between the non-causal drafting blocks and the causal target block that determines the acceptance rate of tokens during generation. At the start of sampling, it would be intuitive that the acceptance rate would be low because the draft model has very little information available to it and so will draft poor quality tokens. However, because the target model is only a single block on top of the non-causal hidden states, the target distribution is also weak in this initial stage. The draft and target distribution could then be aligned even though they are both far from the ground truth predictive distribution at this stage, leading to a high acceptance rate. Without windows, this is non-ideal because generating too many tokens in one go at the start of sampling leads to poor quality performance, indeed standard masking schedules such as cosine typically reveal very few tokens in the first few iterations. We therefore introduced the window functions to limit how many tokens can be sampled from the poor quality target at the start of the generation trajectory.
>
> At the end of the sampling trajectory, the conditioning signal is strong since there are many revealed tokens and so both the draft and the target are good approximations to the true predictive distribution resulting in a high acceptance rate. We therefore also allow many tokens to be revealed at this stage using a large window, finding that this gives the best trade-off between sample quality and NFE.

---

> > ### Comment · Reviewer_fgUG · 2025-11-26
> >
> > Thank the authors for their detailed rebuttal. I will keep my positive score.

---

### Official Review · Reviewer_Joif · 2025-11-01

**Soundness:** 3
**Presentation:** 3
**Contribution:** 3
**Rating:** 4
**Confidence:** 4

**Summary:**

The paper proposes Self-Speculative Masked Diffusions (SSMD), a masked-diffusion generator for discrete data that aims to reduce sampling cost. The key idea is a hybrid transformer: most layers are non-causal to produce a factorized draft, followed by a small causal head that verifies and (via a specialized speculative step) yields a non-factorized distribution over the masked positions in a single forward pass. A theoretical treatment derives a likelihood decomposition for the induced sampler and an ELBO. Empirically, the method reports roughly 2× fewer network forward evaluations (NFE) than a standard MDM at matched sample quality on text (Text8, OpenWebText) and protein sequence generation (UniRef50 + ESMFold pLDDT).

**Strengths:**

1. **Principled modeling & architecture.** The hybrid non-causal/causal stack with σ-aware causal masking is clearly specified and lets the causal head “strictly improve” the draft through a residual connection, aligning draft/target and plausibly raising acceptance rates. The training objective jointly optimizes both heads in a single pass.
2. **Sound theoretical analysis.** The paper characterizes the sampler’s distribution under changing targets and derives a tractable decomposition for likelihood and an ELBO; this is rare for speculative schemes beyond left-to-right AR.
3. **Clear empirical win in NFE.** On OpenWebText, the method matches or improves GPT-2 NLL at **~½ the NFE** relative to a standard MDM baseline, with similar token entropy; ablations support the importance of the residual and “11 non-causal + 1 causal” design.
4. **Protein application with minimal intrusion.** The protein experiment keeps a frozen ESM2-based backbone and trains just the causal head, showcasing plug-in potential.

**Weaknesses:**

1. **Reported “2×” is an NFE metric, not end-to-end latency.** The paper motivates gains by memory-bound inference and single-pass verification, but practical wall-clock speed depends on kernel fusion, batch size, cache behavior, acceptance rates, and framework overheads. The text emphasizes that longer sequences do not necessarily mean worse latency because the cost is memory-bound, yet the experiments still operationalize efficiency largely as NFE. It’s unclear how much of the 2× NFE translates into **real time** speedups on GPUs for moderate batch sizes typical in protein generation.
2. **Protein benchmark strength and comparability.** The protein study uses **average pLDDT from ESMFold** on sequences sampled from UniRef50. While pLDDT is a useful proxy, the field has converged on stronger metric foldability. Compared to "Path Planning for Masked Diffusion Model Sampling", which reports much stronger protein benchmarks with higher-quality samples and broader evaluation, the current results look considerably weaker. This makes it difficult to assess the real progress or competitiveness of this approach in protein generation.

**Questions:**

see weakness.

---

> ### Author Response · Authors · 2025-11-21
> **Response**
>
> We would like to thank the reviewer for their comments and engagement with our work. We really appreciate that our work’s sound theoretical analysis and empirical wins were highlighted in the review. We address your specific questions below.
>
> >Reported “2×” is an NFE metric, not end-to-end latency. ...
>
> In this work we focused on the theoretical characterization of a new class of masked generative models as well as architectural ablations within this model class to understand their impact on sampling quality performance. We therefore focused on reliable and robust measures of computation that generalize across specific hardware, ML frameworks and engineering implementation. This is because modern ML systems are complex and low latency real world deployment of language models requires a large amount of infrastructure and engineering, such as kernel fusion and KV caching as you mention.
>
> Properly and rigorously pushing this engineering effort is out of the scope of this paper as to fairly compare methods on wall clock time, comparable engineering effort needs to be invested into each method. Otherwise, the wall clock time of baselines could be artificially inflated due to being more or less suited to the hardware that the ablation happens to be run on. The metric we use in our paper, NFE, is better for giving an idea of the upper bound achievable speedup as it is robust to implementation differences. Achieving this speedup would then be an engineering question we leave to future work.
>
> > Protein benchmark strength and comparability ...
>
> In this experiment we were not aiming to create a state of the art protein sequence generative model. We aimed to create an experiment that gave us the clearest signal on the benefit of using a non-factorized predictive distribution within a standard mask diffusion model against a factorized predictive distribution. Peng et al. 2025 specifically aim to create a new method for deciding which tokens to unmask, which is an important task in protein sequence generation. They build on the DPLM model (Wang et al. 2024) which includes hand crafted heuristics specifically for protein sequence generation. For example, they use aggressive confidence based orderings which naively leads to sequences with many repeated tokens e.g. “MLKNVVVVVVVVVVLDN”. They then have a threshold to resample repeated tokens in their [code](https://github.com/bytedance/dplm/blob/main/src/byprot/models/dplm/dplm.py). For Peng et al. 2025, their method can be seen as learning the ordering to generate tokens rather than using a confidence based ordering. It therefore makes sense for them to use these elements of the DPLM codebase and push protein sequence generation performance.
>
> However in our case, we focus on the random ordering case, evaluating the impact of the non-factorized versus factorized predictive distribution over all orderings. Including aggressive confidence ordering and sequence resampling heuristics would obscure the signal of the impact of our contribution and hence we focus on the vanilla mask diffusion setup so that we can make generalizable conclusions. We have updated Section 5.3 in the manuscript to comment that better results have been obtained using dedicated ordering selection techniques for protein data and have included a reference to this paper, thanks for pointing this out!
>
> **References**
>
> Fred Zhangzhi Peng, Zachary Bezemek, Sawan Patel, Jarrid Rector-Brooks, Sherwood Yao, Avishek Joey Bose, Alexander Tong, Pranam Chatterjee Path Planning for Masked Diffusion Model Sampling Arxiv 2025 https://arxiv.org/pdf/2502.03540
>
> Xinyou Wang, Zaixiang Zheng, Fei Ye, Dongyu Xue, Shujian Huang, Quanquan Gu. Diffusion Language Models Are Versatile Protein Learners ICML 2024 https://arxiv.org/pdf/2402.18567
>
> DPLM code https://github.com/bytedance/dplm/blob/main/src/byprot/models/dplm/dplm.py  L220

---

> > ### Comment · Reviewer_Joif · 2025-11-26
> >
> > thank you for the response. I'm just concerned about plddt as a synthetic metrics from ESMFold/AlphaFold will be missleading when its to low; as in if all the plddt results are below 60, their relative differences might not be very meaningful and thus hard to tell the effectiveness of the methods. But yes I agree with you and I will raise my score.

---

### Official Review · Reviewer_Fyeq · 2025-11-04

**Soundness:** 3
**Presentation:** 3
**Contribution:** 4
**Rating:** 6
**Confidence:** 4

**Summary:**

This paper proposes self-speculative masked diffusion,  a new class of masked diffusion generative models for discrete data that require significantly fewer function evaluations to generate samples. In self-speculative masked diffusion, the neural network is decomposed into a large bidirectional diffusion model and a small any-order causal autoregressive decoding head. The large bidirectional model produces candidates, while the small decoding head serves as a verifier. Through self-speculative approaches, the authors are able to get a 2x sampling speed boost with barely any quality decrease.

**Strengths:**

* This paper is practically significant. I think overall this technique is very promising for real-world deployment.
* The empirical results are impressive. With barely any loss of generation quality, self-speculative masked diffusions achieve a 2x acceleration, making this almost a free lunch.
* I think overall the paper is well-written, with a clear motivation, a solid method, real innovations, and abundant theoretical analysis.

**Weaknesses:**

* The sampling algorithm is not explained thoroughly enough, especially regarding the attention patterns. Please see Question 3 for more details.
* I think Section 3.4 is less accessible to readers. If the meaning of this proposition can be explained more intuitively, readers will better understand it.

**Questions:**

* In Figure 1, do the dots connecting input tensors mean addition? I am assuming this because the token embeddings and positional encodings are connected by dots, but I think this should be more explicitly indicated.
* For the input to the causal parts, are the hidden states from the non-causal model necessary? I think from a theoretical perspective, only token embeddings and positional embeddings are enough. Can the authors explain, theoretically or empirically, why these hidden states are needed, especially given the residual connections to the output of the causal part?
* My biggest question is, during sampling, how do the authors handle the causal attention? If it still remains a causal structure, if I understand correctly, then in the case of $k\geq 2$, i.e., parallel decoding, the mask tokens in the last are attending to mask tokens before them, leading to a different situation from the model's training scenario. Can the authors provide any justifications on why this does not ruin the generation quality?
* The biggest contribution of this work is to accelerate inference without loss of quality. Can the authors provide any comparison with other acceleration algorithms, such as Fast-dLLM [1]?
* The authors do not point out if any part of the model is frozen during training, so I assume no parts are actually frozen. However, in the protein experiments, the authors freeze the backbone and only trained a causal head. What will the results be if the backbone is also finetuned?

---
**References**

[1] Wu, C., Zhang, H., Xue, S., Liu, Z., Diao, S., Zhu, L., ... & Xie, E. (2025). Fast-dllm: Training-free acceleration of diffusion llm by enabling kv cache and parallel decoding. arXiv preprint arXiv:2505.22618.

---

> ### Author Response · Authors · 2025-11-21
> **Response**
>
> We would like to thank the reviewer for their time and detailed review of our work. We appreciate the comment that our paper is “well-written, with a clear motivation, a solid method, real innovations, and abundant theoretical analysis”. We are happy to address the questions raised in the review below.
>
> >The sampling algorithm is not explained thoroughly enough, especially regarding the attention patterns. Please see Question 3 for more details.
>
> > My biggest question is, during sampling, how do the authors handle the causal attention? ...
>
> Thank you for raising this point of confusion. We agree it would be helpful to mention explicitly the exact inputs into each type of transformer attention block during training and sampling. We have added a paragraph to Section 3.1 clarifying that the non-causal block can see real and mask tokens in the input whereas the causal block only ever sees real tokens.
>
> To describe explicitly what occurs during training and sampling here, we will use a mask diffusion style notation $x_t$ to represent a partially corrupted datapoint that is seen during training/sampling. We will use $x_t^{NC}$ to represent the input to the non-causal transformer and $x_t^{C}$ to represent the input into the causal transformer.
>
> **Training**
>
> Say our real datapoint is made up of 4 tokens: $\text{CAT} \quad \text{IN} \quad \text{THE} \quad \text{HAT}$. We corrupt the datapoint with two mask tokens leading to an input into the non-causal block of
> $$x_t^{NC} = \text{CAT} \quad \mathbf{M} \quad \mathbf{M} \quad \text{HAT}$$
>
> Now, for the causal block, we use the real sequence without mask tokens, but just permuted by some ordering $\sigma$ that is picked at random during training time, in this example we permute the ordering from (1,2,3,4) to (4,1,2,3). This would look like
>
> $$x_t^{C} = \text{HAT} \quad \text{CAT} \quad \text{IN}$$
>
> Note that the mask tokens that were in the non-causal input have now been replaced with their corresponding real token. The final clean token is not included as the causal blocks perform next token prediction and so the final token $\text{THE}$ is predicted from the track corresponding to the token $\text{IN}$ in $x_t^{C}$.
>
> The left-to-right causal mask in the causal block ensures that the correct conditional independencies hold. For the prediction of token $\text{IN}$ the causal block can use the information to the left of $\text{IN}$ i.e. the tokens $\text{HAT}$, $\text{CAT}$ just as the non-causal block can. For prediction of token $\text{THE}$ the causal block can use information to the left of $\text{THE}$ i.e. the tokens $\text{HAT} \quad \text{CAT} \quad \text{IN}$, getting access to the extra token $\text{IN}$ (which is the benefit of using an autoregressive predictive distribution over a non-causal factorized predictive distribution). In our paper in Appendix A we include more detail on the difference between the non-causal and causal attention mask.
>
> **Sampling**
>
> Say we are generating a new datapoint consisting of 4 tokens. Our ordering is such that we are generating in the ordering (4,1,2,3). We have so far generated the 4-th token and 1-st token, being $\text{HAT}$ and $\text{CAT}$ respectively. At the input to the non-causal block we input the non-permuted sequence, with mask tokens to represent positions that have not been generated yet.
>
> $$x_t^{NC} = \text{CAT} \quad \mathbf{M} \quad \mathbf{M} \quad \text{HAT}$$
>
> We now use the factorized predictive distribution to sample draft tokens for all currently unknown positions. Say we sample $\text{ON}$ and $\text{THE}$ for positions 2 and 3 respectively (i.e. our draft model thinks the full sentence is $ \text{CAT} \quad \text{ON} \quad \text{THE} \quad \text{HAT}$). We now place these sampled tokens into the unknown final positions in the permuted sequence. This means the input into the causal block is then
>
> $$x_t^{C} = \text{HAT} \quad \text{CAT} \quad \text{ON}$$
>
> Noting again how the causal block is doing next token prediction so takes one less token than the full sequence as input. The causal block then provides causal probabilities over all drafted positions in parallel which we use to accept/reject the drafted tokens.
>
> In conclusion, we can see that the non-causal blocks see a combination of real and mask tokens whereas the causal blocks only ever see real tokens during both training and sampling so there is no mismatch in terms of the visibility of mask tokens. We note, however, that due to an imperfect model, during sampling time the distribution of input real tokens can be different because the model generates imperfect tokens (e.g. in our example the real sentence was $\text{CAT} \quad \text{IN} \quad \text{THE} \quad \text{HAT}$ but the model tried to generate $\text{CAT} \quad \text{ON} \quad \text{THE} \quad \text{HAT}$). This distribution shift between training and inference is not unique to our method and is a problem for all generative models including autoregressive models and standard MDMs.

---

> ### Author Response · Authors · 2025-11-21
> **Response**
>
> >I think Section 3.4 is less accessible to readers. If the meaning of this proposition can be explained more intuitively, readers will better understand it.
>
> This is a good point and we have added additional intuition in Section 3.4 and around Proposition 3.1.
>
> Covering the details here: to understand the need for Proposition 3.1 in the first place, we need to compare our approach to autoregressive models and MDMs. For an autoregressive model, the log-likelihood that the model generates a certain datapoint is simply the sum of log-likelihoods along the sequence
>
> $$\text{Autoregressive likelihood} \quad \log p(x) = \sum_{d=1}^D \log p(x^d | x^{1:d-1})$$
>
> For an any-order autoregressive model or MDM (assuming a single token is revealed per step), the log-likelihood is bounded by an ELBO based on the ordering of generation
>
> $$\text{Any-order autoregressive likelihood} \quad \log p(x) \geq E_{\sigma} \Big[ \sum_{d=1}^D \log p(x^{\sigma(d)} | x^{\sigma(1:d-1)}) \Big] $$
>
> For our self-speculative approach, neither of these likelihoods apply. The likelihood of our entire algorithm generating a certain datapoint is defined by the non-causal and causal predictive distributions in combination with a self-speculative sampling algorithm. Speculative sampling in this case is not being used as simply a tool to sample from a fixed target distribution but is inherent to the construction of our entire algorithm due to the shifting target.
>
> For autoregressive models and MDMs the above log-likelihoods are good to know because they can be used to understand how the predictive distribution we actually train ($\log p(x^d | x^{1:d-1})$) is related to the likelihood the overall procedure will actually generate a given datapoint. We would also like to characterize how our two predictive distributions (non-causal and causal) come together with the self-speculative algorithm to give an overall likelihood of generating a certain datapoint.
>
> In terms of understanding the actual form of Proposition 3.1, we use dynamic programming ideas which leverage special structure within the problem such that the overall task can be broken up into smaller simpler chunks. In this case, the special structure is the fact that when deciding on whether to accept the token in position $d$: $x^d$, the probability of accepting, $A^d$, is independent of all accept/reject decisions prior to the most recent reject decision. This is because the probability distributions that define the acceptance probability, $ p_{\theta}^{NC} $ and $ p^{C}_{\theta, \phi} $ depend only on the currently revealed tokens and not the accept/reject sequence used to generate those tokens. This suggests that we can split the problem up into smaller chunks by focusing on when rejections occur (the event $R^d$).
>
> Specifically, in our dynamic programming setup, we split our overall likelihood $p(x^{1:D})$ into smaller chunks of the form $p(x^{1:d}, R^d)$. Note here we have removed reference to $\sigma$ relative to Proposition 3.1 to keep things simple. The likelihood of the smaller chunk can be written as a linear sum of previous small chunks
>
> $$p(x^{1:d}, R^d) = \sum_{k=1}^d p(x^{1:k-1}, R^{k-1}) p(x^{k:d}, A^{k:d-1}, R^d | x^{1:k-1}, R^{k-1})$$
>
> Using this recursion, we can build up the values of $p(x^{1:d}, R^d)$ for all values of $d$ up to $D$. Then we can write the solution to the overall problem as a sum of smaller chunks that we have already computed
>
> $$p(x^{1:D}) = p(x^{1:D}, A^{1:D}) + \sum_{d=1}^D p(x^{1:d}, R^d) p(x^{d+1:D}, A^{d+1:D} | x^{1:d}, R^d)$$
>
> > In Figure 1, do the dots connecting input tensors mean addition? I am assuming this because the token embeddings and positional encodings are connected by dots, but I think this should be more explicitly indicated.
>
> The dots in Figure 1 represent the concatenation of input tensors along the channel dimension. We included this in a legend in Figure 1 and we have now added this to the figure caption as well to make sure this is explicit. Thanks for pointing this out!

---

> ### Author Response · Authors · 2025-11-21
> **Response**
>
> >For the input to the causal parts, are the hidden states from the non-causal model necessary? ...
>
> You are indeed right that theoretically the token and positional embeddings are enough to define a causal distribution. However, not passing the hidden states from the non-causal blocks to the causal blocks is effectively wasting computation that has been spent on creating useful representations that can initialize the causal block operations. When these hidden states are passed through, we are effectively turning a 1 causal block target distribution into an 11 + 1 block target model.
>
> This is important especially for a causal any-order block because pure causal transformers perform poorly when applied to any-order sequence generation. For example, in Table 1 in Hoogeboom et al. 2022, a 12 block any-order transformer with a causal mask has a significantly worse negative log-likelihood on text8 (1.64) compared to the mask diffusion style OA-ARDM method that obtains a 1.43 NLL. If we have a single causal block we would expect even worse performance. Even if we add on the non-causal representations at the output of the causal block, if the causal block is much worse than the non-causal blocks, it cannot add on any extra useful information. Furthermore, since non-causal blocks are more effective at processing any-order information (as can be seen in Table 1 in Hoogeboom et al. 2022) we mitigate the weaknesses of any-order causal blocks by allowing it to re-use the computations carried out by the non-causal blocks.
>
> We note that the re-use of hidden states in self-speculative style architectures is not new and is done, for instance, in Cai et al. 2024 the hidden states from the pre-trained LLM are used in the prediction heads even though there is also a residual connection at the output of the head with the original hidden state.
>
> **References**
>
> Emiel Hoogeboom, Alexey A. Gritsenko, Jasmijn Bastings, Ben Poole, Rianne van den Berg, Tim Salimans. Autoregressive Diffusion Models ICLR 2022  https://arxiv.org/pdf/2110.02037
>
> Tianle Cai, Yuhong Li, Zhengyang Geng, Hongwu Peng, Jason D. Lee, Deming Chen, Tri Dao. MEDUSA: Simple LLM Inference Acceleration Framework with Multiple Decoding Heads ICML 2024 https://arxiv.org/pdf/2401.10774

---

> ### Author Response · Authors · 2025-11-21
> **Response**
>
> > The biggest contribution of this work is to accelerate inference without loss of quality. Can the authors provide any comparison with other acceleration algorithms, such as Fast-dLLM [1]?
>
> In Fast-dLLM, acceleration of a vanilla mask diffusion model is achieved through selection of a specific decoding ordering that maximizes model confidence. The argument is made that as long as positions of sufficient confidence are selected, the error incurred by the factorized approximation is limited because both the true non-factorized probability and the approximate factorized probability for a given token are close to 1 (Theorem 1 in Wu et al. 2025).
>
> This argument is valid only up to a certain number of tokens to be revealed in one step because, fundamentally, approximating a non-factorized distribution with a factorized distribution will always incur error. We can perform a worst case analysis on the dependence of this error on the number of tokens we attempt to reveal in one step by appealing to arguments made in Lavenant and Zanella 2025 and Li and Cai 2025. We will find that the worst case error grows with the number of tokens revealed in one go thus setting a limit on the speedups achievable by factorized methods. Our method on the other hand samples from a non-factorized distribution and is thus free of such limits.
>
> Following the decomposition of Lavenant and Zanella 2025, the KL divergence between the true distribution $\pi(x)$ and the distribution $p_{\text{alg}}(x)$ generated by the sampling algorithm with a factorized predictive distribution can be split into two error terms, one relating with how well our model approximates the true distribution and the other relating to our factorized approximation.
>
> $$\\text{KL}( \\pi(x) || p_{\\text{alg}}(x) ) \\leq E_{\\text{learn}} + E_{\\text{fact}}$$
>
> The factorization error can be written in terms of a sum of total correlations that measure the discrepancy between the non-factorized true distribution and the factorized approximation
>
> $$E\_{\\text{fact}} = \\mathbb{E}_{\pi(x)} \big[ \sum\_{i = 0, k, 2k, \dots}  \\text{TC}\_{\\pi}^{k,i}  \big]$$
>
> where
>
> $$\text{TC}_{\pi}^{k,i} =  \\mathbb{E}\_{\pi(x)} \big[   \log \frac{ \pi(x^{\sigma(i+1:i+k)} | x^{\sigma(1:i)}) }{ \prod\_{d=i+1}^{i+k} \pi(x^{\sigma(d)} | x^{\sigma(1:i)})  } \big] $$
>
> We now find a worst case upper bound on $TC_{\pi}^{k,i}$
>
> $$\text{TC}\_{\pi}^{k,i} = \sum\_{d=i+2}^{i+k} \mathbb{E}\_{\pi(x)} \left[ \log \frac{\pi(x^{\sigma(d)} | x^{\sigma(1:d-1)})}{\pi(x^{\sigma(d)} | x^{\sigma(1:i))})} \right]$$
>
> $$\leq \sum_{d=i+2}^{i+k} \mathbb{E}_{\pi(x)} \left[ \log \frac{\pi(x^{\sigma(d)} | x^{-\sigma(d)})}{\pi(x^{\sigma(d)})}\right]$$
>
> $$\leq \sum_{d=i+2}^{i+k} - \mathbb{E}_{\pi(x)} \left[ \log \pi(x^{\sigma(d)}) \right]$$
>
> $$\leq (k-1) \log | \mathcal{X} |$$
>
> where on the second line we have used monotonicity of the KL with respect to the conditioning, more conditioning on one side and less conditioning on the other can only increase the KL. On the fourth line we have used the fact the entropy of $\pi(x^{\sigma(d)})$ is less than the entropy of a uniform distribution over the state space. We use $x^{-\sigma(d)}$ to represent all the dimensions of $x$ other than $\sigma(d)$ and $|\mathcal{X}|$ to represent the size of the state space of $x$.
>
> In contrast to this result, in Fast-dLLM Theorem 1-(2) a TV bound is obtained between the true distribution $p$ and the factorized approximation $q$ of the form
>
> $$D\_{TV}(p, q) < (3n -1)/2 \epsilon$$
>
> where $n$ is the number of tokens revealed in a single step and epsilon is the level of confidence
>
> $$p(x^{\sigma(d)} | x^{\sigma(1:i)}) = 1 - \epsilon$$
>
> We can see that their bound also grows with the number of tokens revealed in one go and so they need $\epsilon$ to be small (a very confident model) in order to counteract the factorized model approximation error. $\epsilon$ cannot always be small when sampling from a non-factorized joint distribution simply from the fact that if $\epsilon$ was always close to 0 then the joint distribution would be close to totally factorized which is clearly not the case with real data.
>
> In our method, we circumvent these difficulties with factorized approximation by sampling from a non-factorized predictive distribution.
>
> **References**
>
> Chengyue Wu, Hao Zhang, Shuchen Xue, Zhijian Liu, Shizhe Diao, Ligeng Zhu, Ping Luo, Song Han, Enze Xie Fast-dLLM: Training-free Acceleration of Diffusion LLM by Enabling KV Cache and Parallel Decoding. Arxiv 2025 https://arxiv.org/pdf/2505.22618
>
> Hugo Lavenant, Giacomo Zanella. Error Bounds and Optimal Schedules for Masked Diffusions with Factorized Approximations. Arxiv 2025 https://www.arxiv.org/pdf/2510.25544
>
> Gen Li, Changxiao Cai. A Convergence Theory for Diffusion Language Models: An Information-Theoretic Perspective. Arxiv 2025 https://arxiv.org/pdf/2505.21400

---

> ### Author Response · Authors · 2025-11-21
> **Response**
>
> > The authors do not point out if any part of the model is frozen during training, so I assume no parts are actually frozen. However, in the protein experiments, the authors freeze the backbone and only trained a causal head. What will the results be if the backbone is also finetuned?
>
> Thank you for pointing out this lack of clarity, we have updated Section 3.2 in the main text to make clear that our method is compatible with both full training and frozen backbone training. We would like to emphasize that in our text experiments we investigate full training of the backbone alongside the causal blocks to cover the case that our method is implemented by users with access to large amounts of training resources.
>
> On the other hand, in our protein experiments, we investigate the compute limited regime wherein the user has access to limited GPUs and can only fine-tune the head on top of the frozen backbone. Indeed, this experiment was completed on only a single A100 GPU. In order to run full finetuning, hyperparameters would need to be selected carefully such as a longer warmup for the backbone parameters whilst the causal head is training to avoid unstable signal passing through the causal head to the non-causal blocks in the early stages of fine-tuning. For example these differential learning rates are used by Cai et al. 2024 when training MEDUSA heads. Due to these extra complications, we leave investigation of full fine-tuning in the compute unconstrained setting to future work.
>
> **References**
>
> Tianle Cai, Yuhong Li, Zhengyang Geng, Hongwu Peng, Jason D. Lee, Deming Chen, Tri Dao. MEDUSA: Simple LLM Inference Acceleration Framework with Multiple Decoding Heads ICML 2024 https://arxiv.org/pdf/2401.10774

---

### Author Response · Authors · 2025-11-21
**Response to Reviewers**

We would like to thank the reviewers for their constructive feedback and time in preparing their reviews. We were especially pleased that our paper was considered to present a “solid and clever contribution” (reviewer FGUG) with a “clear motivation”, “real innovations” and “abundant theoretical analysis” (reviewer FYEQ).

Based on the reviewer’s comments we have updated the manuscript with new additions being highlighted in blue. We have additionally addressed each of the reviewer’s questions individually below.

---

### Meta-Review · Area_Chair_kMvU · 2025-12-26

**Summary:**

The paper introduces an architecture, training loss, and inference procedure for a model that is both trained "causally" and "non-causally", that allows one to introduce a "self-speculation" approach in which the inference procedure proposes a chunk of generation "any-order autoregressively/causally" and is then is verified via a "rejection-sampling" style approach in parallel.

The main contribution in my view is the architecture --- the training loss is sort of a straightforward "mixture" of the causal and non-causal loss, and the speculative decoding inference-time procedure largely follows the standard techniques from speculative decoding in standard, autoregressive setups. There is some theory to show that the rejection sampling is "set up right", and to analyze the number of rejection steps --- though this is fairly standard for readers with standard MCMC background. Empirically, the approach seems quite promising, getting roughly 2x NFE savings, with very little loss in accuracy.

The reviewers were largely positive, with reviewer Joif promising to update their score before the Openreview incident. The concerns of the reviewers include whether NFE is a reasonable metric (which I am sympathetic towards --- though the authors replied that wall-clock time is succeptible to hardware optimizations, which is also true). Another concern was clarity of writing (Fyeq), which the edits somewhat ameliorated --- but in my opinion, the appendix is still somewhat messy. (I think for instance, the rejection sampling analysis is quite tedious, and some improvements in notation would likely make the writing cleaner.)

**Reviewer Concerns:**

NFEs as a metric of performance, meaningfulness of protein benchmark, sensitivity to new hyperparameters.

**Reviewer Scores:**

Reviewer Joif said explicitly they'd raise their score (the lowest score, a 4). I believe that some of the writing concerns of reviewer Fyeq were addressed.

---

### Decision · Program_Chairs · 2026-01-26

Accept (Poster)